# REAL-TIME EVALUATION FOR NOVEL CLASS DISCOVERY AT TEST TIME

## ABSTRACT

We introduce Test-Time Discovery (TTD), a real-time evaluation protocol for novel class discovery under sequential test-time conditions. Unlike post-hoc Novel Category Discovery (NCD) evaluation, which assesses clustering only after the full test set is processed, TTD requires models to classify known categories and discover novel ones in real time as samples arrive. To address this setting, we propose a training-free Hash Memory (HM) method. HM encodes feature norm and direction into semantic-aware hash codes, enabling locality-sensitive hashing for efficient retrieval and consistent reuse of discovered classes. A global-to-local strategy combines prototypes for stable known-class predictions with memory-based reasoning for flexible novel discovery. A lightweight self-correction mechanism further improves reliability by removing mislabeled entries from early discoveries. Experiments on diverse benchmarks show that HM achieves more accurate and stable real-time discovery than NCD methods, while maintaining performance on known classes. Our code will be released.

## 1 INTRODUCTION

Recent progress in Novel Category Discovery (NCD) (Zhong et al., 2021; Vaze et al., 2022) has shown that models can group previously unseen categories without supervision. *However, evaluation is usually **post-hoc**, performed by reclustering the entire test set as a separate process. As a result, discovery itself is only optimized for the final clustering outcome, rather than being evaluated during the sequential test-time process.* In parallel, test-time training (Wang et al., 2020; Boudiaf et al., 2022) adopts real-time evaluation during deployment, but it focuses exclusively on distribution shifts among predefined categories without addressing novel class discovery.

Inspired by test-time training, we propose a framework that performs *test-time prediction and novel class discovery simultaneously*, enabling a model trained on known classes to operate in open environments and support real-time evaluation as new samples arrive. We refer to this NCD extension as **Test-Time Discovery (TTD)** (Fig. 1), which emphasizes not only discovering novel categories but also evaluating prediction quality in real time. Unlike the post-hoc evaluation in NCD, which assesses clustering quality after the discovery process is complete, many real applications require timely discovery and immediate prediction during deployment, such as detecting unexpected obstacles in autonomous driving. In TTD, the model must not only classify known categories but also decide in real time whether a sample belongs to a previously discovered class or should initiate a new one, a requirement that fundamentally differs from NCD, where all test data are available at once and decisions are made only after global clustering.

Some recent NCD variants, such as on-the-fly category discovery (Du et al., 2023), also support online discovery. However, because the evaluation is still post-hoc, these methods are designed to optimize final clustering quality, *rather than ensuring that each test-time sample is correctly discovered or classified as it arrives*. As shown in Fig. 1, once we shift the focus to the correctness of discovery at test time, three unique **challenges** emerge: (1) distinguishing truly novel classes from already discovered ones, (2) coping with sparse evidence when a new class first appears, and (3) maintaining stability on known classes to avoid forgetting. Existing NCD and test-time training methods often rely on confidence thresholding, which easily leads to redundant discoveries and error accumulation, underscoring the need for more reliable methods under our evaluation protocol.

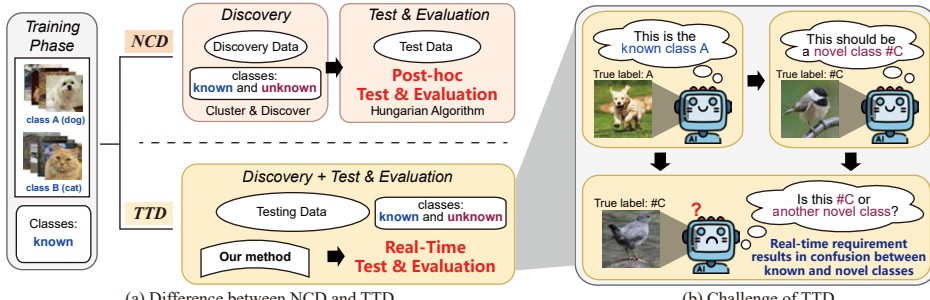

(a) Difference between NCD and TTD  (b) Challenge of TTD

Figure 1: Test-Time Discovery (TTD). A model trained on known classes is deployed in an open environment where test samples arrive sequentially. Unlike post-hoc NCD, TTD requires *real-time evaluation*, deciding for each sample whether it belongs to a known or discovered class or should initiate a new one, which suffers from confusion between known and novel classes.

In this paper, we propose a training-free Hash Memory (HM) method tailored for TTD. At its core, HM leverages semantic-aware hashing to encode both the scale and direction of test features, enabling efficient retrieval of geometrically similar instances and reducing redundant rediscoveries of already identified classes. To ensure reliable classification under sequential conditions, HM further adopts a collaborative strategy in which confident predictions are handled by global prototypes, while ambiguous cases are resolved through memory-based reasoning, thereby stabilizing performance on known categories while flexibly accommodating new ones. Finally, because early discoveries often assign unreliable pseudo labels when a novel class appears with few samples, we introduce a lightweight self-correction mechanism that periodically revisits memory entries as more evidence arrives and removes mislabeled samples, improving the robustness of discovery. Through these designs, HM provides a practical and effective solution for real-time novel class discovery, maintaining accuracy on known classes while delivering consistent gains over existing NCD and test-time training approaches. Our contributions are three-fold:

(1) We introduce TTD, a real-time evaluation protocol for novel class discovery under test-time conditions. Unlike post-hoc NCD evaluation, TTD requires models to recognize known classes and discover novel classes on the fly, aligning evaluation more closely with real-world scenarios.

(2) We propose a training-free hash-based memory method tailored for TTD, which integrates semantic-aware hashing, collaborative classification, and self-correction to address the unique challenges of real-time discovery.

(3) Through experiments on diverse benchmarks, we show that our approach outperforms existing discovery and test-time training methods in real-time discovery while maintaining stability on known classes, providing a strong baseline for future research under the TTD protocol.

## 2 RELATED WORK

**Novel Category Discovery (NCD).** NCD (Han et al., 2019; Zhang et al., 2023; Cendra et al., 2025) addresses the problem of identifying and grouping samples from unseen classes without supervision. Typical approaches rely on representation learning followed by clustering. This framework has enabled progress in learning transferable features and discovering novel semantic structures, but the post-hoc nature of evaluation means that discovery quality is only measured after the process is complete, rather than during deployment. Generalized Category Discovery (GCD) (Vaze et al., 2022; Pu et al., 2023) extends NCD by requiring recognition of both known and unknown categories, but it assumes full access to the entire dataset and relies on global clustering. Ccontinual GCD (CGCD) (Cendra et al., 2025) extends GCD to incremental streams of novel classes, yet evaluation still remains post-hoc. On-the-fly Category Discovery (OCD) (Du et al., 2023; Zheng et al., 2024) attempts to move closer to deployment settings by classifying both known and novel categories at test time. However, all NCD, GCD, CGCD and OCD are still primarily benchmarked in offline settings, where clustering quality is measured after discovery rather than in real time, leaving open the question of how to evaluate models that must provide predictions on the fly.

**Test-Time Training.** Test-time training (Sun et al., 2020; Liu et al., 2021; Gandelsman et al., 2022; Osowiechi et al., 2023; Su et al., 2024) refers to updating model parameters at inference by optimizing auxiliary self-supervised objectives such as masked reconstruction and contrastive consistency. These objectives exploit the structure of incoming test samples, enabling the model to adapt its feature extractor without access to source data. Test-time adaptation (TTA) (Wang et al., 2020; Liang et al.,

2020; Boudiaf et al., 2022; Chen et al., 2022) instead focuses on lightweight parameter updates, such as adapting batch normalization statistics or entropy minimization, and has been shown to improve robustness in scenarios with mild distribution shifts. Both paradigms emphasize domain adaptation under test-time conditions, where the label space remains fixed. However, in open-world deployment, the label space may expand, and models must go beyond adapting to shifted distributions: they must also detect and categorize novel classes as they appear. This highlights the need for evaluation settings that simultaneously consider distributional robustness and novel-class discovery.

# 3 TEST-TIME DISCOVERY

## 3.1 TTD: PROBLEM DEFINITION AND THE CHALLENGE

**Problem Definition**. Following prior works in NCD, we categorize test samples into three groups: *known*, *seen*, and *unseen*. "Known" refers to the training categories $\mathcal{Y}_{known}$, "seen" denotes novel classes that have already been discovered and assigned a new label during testing, and "unseen" indicates novel classes that have not yet been discovered. Formally, the model is trained on $\mathcal{D}_{known}^{train} = (\mathcal{X}_{known}, \mathcal{Y}_{known})$, and evaluated on a test stream $\mathcal{D}^{test}$ that contains both known and unknown classes without labels. At test time, the model must (1) classify samples from $\mathcal{Y}_{known} \cup \mathcal{Y}_{seen}$, and (2) initiate new categories for samples from $\mathcal{Y}_{unseen}$.

**Setting Comparison**. Table 1 compares TTD with related discovery settings through their evaluation protocols. Unlike NCD, GCD, CGCD, and OCD, which perform discovery and evaluation *post-hoc* using the full test set, TTD requires models to make irreversible decisions on a streaming test sequence, classifying known classes and initiating novel ones as samples arrive. This distinction removes access to future samples and prevents global reclustering, revealing challenges that post-hoc settings do not capture. TTD addresses this gap by jointly measuring discovery and classification during testing.

Table 1: Comparison of evaluation protocols. K: known classes, U: unknown classes.

| Task | Train | Test | Discovery | Evaluation |
|------|-------|------|-----------|-----------|
| NCD | K | U | Post-hoc | Post-hoc |
| GCD | K | K + U | Post-hoc | Post-hoc |
| CGCD | K | K + $U_1$ + $U_2$ + $\cdots$ | Post-hoc | Post-hoc |
| OCD | K | K + U | Real-time | Post-hoc |
| TTA | K | Shifted K | N/A | Real-time |
| TTD | K | K + U | Real-time | Real-time |

**Challenges**. TTD requires real-time evaluation, making it more demanding than conventional NCD. The main challenges are threefold: *(1) distinguishing novel class discovery from the identification of already discovered ones, (2) coping with the scarcity of samples when a novel class first appears*, and *(3) maintaining performance on known classes while incorporating new ones to avoid forgetting (Kirkpatrick et al., 2017)*. To illustrate these difficulties, we conduct a toy TTD experiment on MNIST using the classical OCD method PHE (Zheng et al., 2024) with thresholding for novelty detection.

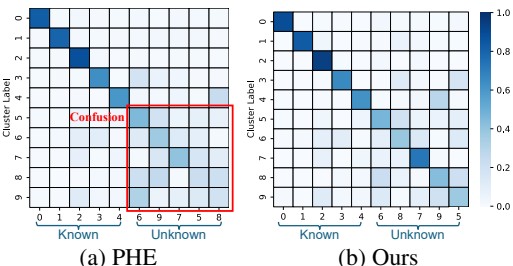

(a) PHE          (b) Ours

Figure 2: Class-cluster prediction matching matrix on MNIST (5K+5U), showing how existing OCD method PHE leads to confusion in TTD.

As shown in Fig. 2a, although clustering visualization appears reasonable, the confusion matrices reveal severe errors: novel classes are repeatedly rediscovered due to unclear boundaries, and previously discovered categories are often misclassified as new. This highlights how chasing post-hoc performance in OCD masks such issues, whereas real-time evaluation in TTD exposes them directly.

# 4 METHOD

## 4.1 OVERVIEW: A TRAINING-FREE FRAMEWORK

We propose a training-free framework called **Hash Memory (HM)** for TTD, where model parameters remain fixed and incoming features are organized in a hash-based memory for online, non-revisitable decisions. HM comprises three components tailored to TTD: (1) a semantic-aware hash that encodes feature norm and direction for efficient retrieval; (2) a collaborative classifier combining global prototypes with local hash-based neighbors; and (3) a lightweight self-correction module that revises early assignments as evidence accumulates. Together, they form a unified pipeline enabling real-time novel-class discovery while preserving known-class performance.

## 4.2 LOCAL-SENSITIVELY HASHING MEMORY

A central difficulty in TTD is distinguishing truly novel classes from those that have already been discovered. Existing NCD methods often fail, repeatedly triggering redundant discoveries due to ambiguous decision boundaries. To address this, we maintain a memory buffer of past test samples, allowing the model to compare each incoming instance with previously discovered categories before creating a new one. To make such retrieval efficient under sequential test-time constraints, we design semantic-aware hash codes and organize them into buckets using Locality-Sensitive Hashing (LSH), which enables fast neighbor search without exhaustive comparisons. Note that the hashing module is not introduced as a new hashing technique but simply as a lightweight mechanism for organizing memory for efficient lookup in streaming TTD.

**Semantic-aware hashing via feature norm and direction**. While memory enables comparisons with past samples, an effective representation is required to decide whether two instances belong to the same category. Prior studies suggest that novel-class features typically have smaller norms and larger directional variance than known classes (Dhamija et al., 2018; Park et al., 2023). Motivated by this, we design a hashing function that jointly encodes the norm and direction of a test sample $x$:

$$h(x) = \left[ \lfloor \kappa \| f(x) \| \rfloor, \mathbf{1}(f(x)^\top \mathbf{r}_1), \cdots, \mathbf{1}(f(x)^\top \mathbf{r}_n) \right], \tag{1}$$

where $\kappa$ is a discretization coefficient that quantizes the feature norm into bins ($\lfloor \kappa \| f(x) \| \rfloor$), ensuring a compact hash space. The remaining $n$ binary terms capture directional information by recording the sign of inner products between feature $f(x)$ and random directions $\{\mathbf{r}_1, \cdots, \mathbf{r}_n\}$. Each unique hash code defines a **bucket** $\mathcal{B}$, which groups together samples with similar norm–direction properties. Organizing the memory in this way enables efficient neighbor retrieval and consistent identification of previously discovered novel classes, thereby reducing redundant rediscoveries.

**Constructing memory buffer when testing**. To support sequential discovery, we maintain memory buffers that store a limited number of representative features for each class, with capacity $M$ samples per class. Two types of buffers are used: (1) a fixed buffer $\mathcal{M}_{\text{known}}$ initialized from training data for all known classes, and (2) a dynamic buffer $\mathcal{M}_{\text{unknown}}$ that is expanded online as new categories are discovered. For a given class $c$, the memory is defined as

$$\mathcal{M}_c = \{(h(x_1), f(x_1)), (h(x_2), f(x_2)), \cdots\}, \tag{2}$$

where each entry stores both the semantic-aware hash code $h(x)$ and the corresponding feature $f(x)$. The hash codes partition the representation space into buckets, so that samples with similar norm–direction properties are assigned to the same bucket. Given a test instance $x$, we query the memory by retrieving its bucket:

$$\mathcal{B}(x) = \{(f(x'), y') \mid h(x') = h(x)\}, \tag{3}$$

where $y'$ denotes the (pseudo-)label of a stored sample $x'$. This bucket-level retrieval ensures that the model only compares $x$ with semantically similar past instances, making class identification more efficient and reliable. During testing, the known-class buffer $\mathcal{M}_{\text{known}}$ remains fixed. When a novel class is first detected, a new buffer $\mathcal{M}_{\#c}$ (where $\#c$ denotes the index of a newly discovered category) is created in $\mathcal{M}_{\text{unknown}}$, and further samples assigned to this class update the buffer. Since testing proceeds sequentially, we adopt reservoir sampling (Vitter, 1985) to maintain representative coverage under limited capacity $M$. Nevertheless, some entries may contain noisy pseudo-labels due to early discovery errors. We address this issue in Sec. 4.4.

## 4.3 PREDICTION AND DISCOVERY BASED ON LSH

Given the hashing memory, prediction and discovery at test time proceed in three stages: (1) an LSH-based classifier uses bucket-level retrieval to distinguish existing from novel classes; (2) graph neighbor searching enhances robustness under sparse or noisy buckets; and (3) a global-to-local strategy stabilizes known-class predictions while preserving flexibility for novel discovery.

**Bucket-based prediction and discovery**. When a new test sample arrives in TTD, the model must decide whether it extends an existing discovered category or signals the emergence of a truly novel one. To achieve this, we leverage the semantic-aware hash codes from Eq. (1) not merely as an efficiency tool, but as a principled way to partition the representation space into stable buckets that anchor class identities across time. Given a test sample $x$, we query its bucket $\mathcal{B}(x)$ and classify it by:

$$y = \begin{cases} y_{\text{vote}}, & \text{if } \mathcal{B}(x) \neq \emptyset, \\ y_{\text{new}}, & \text{otherwise,} \end{cases} \tag{4}$$

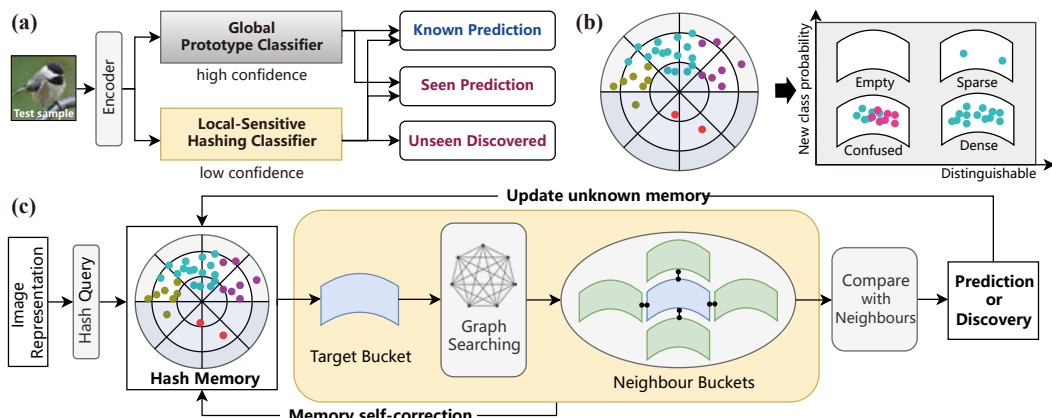

Figure 3: Overview of the Proposed Method. (a) High-confidence samples are predicted via a prototype classifier, and others are routed to the LSH-based classifier. (b) Samples are stored in buckets indexed by feature norm and direction, and sparse buckets indicate potential novel classes. (c) A test sample is hashed to locate its primary bucket. If no similar instance is found, the sample is flagged as novel. Otherwise, it inherits the most relevant neighbor label.

where $y_{\text{vote}} = \arg\max_{y \in \mathcal{Y}_{\text{test}}} \sum_{(f(x'),y') \in \mathcal{B}(x)} \mathbb{I}(y' = y)$. If multiple classes tie, the one with the smallest average distance to $x$ is chosen. If $\mathcal{B}(x)$ is empty or too sparse, the sample is regarded as a new class and assigned a temporary label such as "#$c$" before being added to memory. This design ensures that once a category has been discovered, subsequent samples can reuse it via bucket-level retrieval, thereby reducing redundant rediscoveries. It also helps stabilize predictions across known and novel classes, but relying on a single bucket can still be sensitive to noise or sparsity.

**Graph-augmented retrieval for robust discovery.** While bucket-level retrieval reduces redundant rediscoveries, it can be unreliable when data are noisy or sparse. This issue is critical in TTD, where novel classes often start with only a few samples. To enhance stability, we construct a dynamic graph in which each bucket is treated as a node and connected to its most similar neighbors based on the cosine similarity between their mean feature directions. As the memory buffer receives new samples, the mean direction of each bucket is updated, and the similarities are refreshed accordingly, yielding an event-driven, lightweight graph update. For a test sample $x$, we expand its candidate set by retrieving the top-$k$ neighboring buckets $\mathcal{B}'_1(x), \ldots, \mathcal{B}'_k(x)$ and form an augmented bucket:

$$\hat{\mathcal{B}}(x) = \mathcal{B}(x) \cup \mathcal{B}'_1(x) \cup \cdots \cup \mathcal{B}'_k(x).$$

The LSH-based classifier is then applied to $\hat{\mathcal{B}}(x)$, leveraging both local evidence and supportive context from related neighbors. This graph-augmented retrieval mitigates the instability of sparse buckets and improves discovery of novel classes under TTD.

**Global-local hybrid classifier**. The LSH-based classifier is effective for ambiguous or novel categories, but relying only on localized retrieval can fragment semantically close known classes. In contrast, a global prototype classifier provides stable separation of known categories but lacks flexibility for novel discovery. We therefore adopt a hybrid strategy: each test sample is first evaluated by the prototype classifier, and if the cosine-similarity confidence is high, it is assigned to the corresponding known class; otherwise, the LSH-based classifier (Eq. 4) refines the decision using bucket neighbors and may assign a novel label. This simple mechanism preserves stability on known classes while allowing flexible discovery of new ones. Since the global classifier depends on prototypes, they are updated online: for a seen class, we apply exponential moving average (EMA) with factor $\alpha$, and for a new class, the first observed feature serves as its prototype:

$$\mu_c = \begin{cases} \alpha\mu_c + (1 - \alpha)f(x), & \text{if } c \in \mathcal{Y}_{\text{seen}}, \\ f(x), & \text{if } c \notin \mathcal{Y}_{\text{test}}. \end{cases} \tag{5}$$

Known prototypes are kept fixed to avoid forgetting, which is similar with many practices in few-shot learning (Hajimiri et al., 2023; Sakai et al., 2025).

### 4.4 MEMORY SELF-CORRECTION MECHANISM

The reliability of hash memory is critical in TTD, since pseudo labels assigned during early discovery are often unreliable. When novel classes first emerge with very few samples, the model may

mistakenly treat erroneous instances as new categories and store them in memory, which can mislead subsequent predictions. To mitigate this issue, we introduce a task-specific *self-correction (SC) mechanism* that is tightly coupled with the LSH-based memory design.

Unlike generic pseudo-label refinement, SC operates locally within the hash memory. It periodically selects a small fraction (less than 10%) of stored samples from each novel class and re-evaluates their labels by voting consistency with their bucket neighbors, extended through graph-augmented retrieval (Sec. 4.2). This process corrects mislabeled instances while avoiding full-scale memory refresh. Formally, for a memory entry $(f(x), y) \in \mathcal{M}_{\text{unknown}}$, its label is reassigned as

$$y(x) = \arg\max_{y \in \mathcal{Y}} \sum\nolimits_{(f(x'),y') \in \hat{\mathcal{B}}(x)} \mathbb{I}(y' = y), \qquad (6)$$

where $\hat{\mathcal{B}}(x)$ denotes the augmented bucket of $x$. If $y(x) \in \mathcal{Y}_{\text{known}}$, the entry is discarded, and future test samples will replenish the buffer, ensuring long-term quality under limited capacity. This mechanism improves cluster purity by correcting mislabeled instances, and it maintains representative coverage by replacing discarded entries with fresh ones.

### 4.5 THE ALGORITHM

Algorithm 1 summarizes the pipeline of HM. Given a test sample $x$, its hash code is first computed using Eq. (1) to locate the corresponding bucket $\mathcal{B}(x)$, which is further expanded into an augmented bucket $\hat{\mathcal{B}}(x)$ through graph neighbor retrieval. Prediction then follows a global-to-local strategy: if the global prototype classifier produces a high-confidence match, the sample is classified as a known class; otherwise, the LSH-based voting in Eq. (4) is applied on $\hat{\mathcal{B}}(x)$. If no reliable neighbors are found, $x$ is regarded as a new class and assigned a temporary label. Finally, prototypes are updated for discovered classes (Eq. (5)), and the memory buffer is maintained with self-correction (Sec. 4.4) to refine noisy entries. This process enables HM to jointly achieve classification of known categories, discovery of novel ones, and correction of early discovery errors in a unified framework.

---

**Algorithm 1** Hash Memory for TTD

1: **Input:** input $x$, boundary $\epsilon$, prototypes $\mu_{c \in \mathcal{Y}_{\text{test}}}$
2: Compute hash value $h(x)$ using Eq. equation 1
3: Search target bucket $\mathcal{B}(x)$ via the hash value
4: Graph searching for neighboring joint bucket $\hat{\mathcal{B}}(x)$
5: Compute confidence $u$ via prototype comparison
6: **if** $u > \epsilon$ **then**
7:     $y = \arg\max_{c \in \mathcal{Y}_{\text{test}}} \frac{f(x)^{\top} \mu_c}{\|f(x)\|\|\mu_c\|}$
8: **else if** $\hat{\mathcal{B}}(x) \neq \emptyset$ **then**
9:     $y = \arg\max_{y \in \mathcal{Y}_{\text{test}}} \sum_{(f(x'),y') \in \mathcal{B}(x)} \mathbb{I}(y' = y)$
10: **else**
11:     $y = y_{\text{new}}$              *# Novel class discovered*
12: **end if**
13: Update prototypes via Eq. equation 5
14: Update and self-correct memory buffer
15: **Output:** prediction $y$

---

## 5 EXPERIMENT

### 5.1 EXPERIMENTAL DETAILS

**Dataset construction**. We conduct our experiments based on four benchmark datasets. CIFAR100 (C100)(Krizhevsky et al., 2009) is a commonly used classic dataset. Tiny ImageNet(Le & Yang, 2015) is a subset of ImageNet(Deng et al., 2009). Caltech-UCSD Birds-200-2011 (CUB)(Wah et al., 2011) and FGVC-Aircraft(Maji et al., 2013) are complex fine-grained dataset. As shown in Table 2, all these datasets are split into known and unknown classes (7:3). The model is trained on the known training set and tested on the

Table 2: Known and unknown class counts of datasets.

| Dataset | #K | #U |
|---|---|---|
| CIFAR100D | 70 | 30 |
| CUB-200D | 140 | 60 |
| Tiny-ImageNetD | 140 | 60 |
| AircraftD | 70 | 30 |

mixture of known and unknown test sets. Because the dataset is used for discovery, we name the four transformed datasets CIFAR100D, CUB-200D, Tiny-ImageNetD, and AircraftD. The More dataset details can be seen in Appendix B.

**Implementation details**. We build our method on the prompt-based method L2P (Wang et al., 2022b), which employs a ViT-B/16 backbone following (Cendra et al., 2025). To improve the representation, when we fine-tune the retained model on the known classes, we follow objectosphere (Dhamija et al., 2018) to reduce the norm in the loss function.

**Evaluation metrics**. Our primary focus is on *real-time evaluation*, and we mainly report the final accumulated value. The evaluations consider known and novel classes. For known classes, we adopt standard accuracy (**KA**) and accuracy forgetting (**KF**). For novel classes, the predicted

Table 3: Comparisons on CIFAR100D, CUB-200D, and Tiny-ImageNetD, AircraftD. Real-time evaluation reflects the accumulated performance across all test batches, while post evaluation reassesses all test samples after testing. **Bold** and underlining denote the best and second-best results.

| | Method | Real-time Evaluation | | | Post Evaluation | | | | | | | |
|---|---|---|---|---|---|---|---|---|---|---|---|---|
| | | KA↑ | TA↑ | CA↑ | KA↑ | TA↑ | CA↑ | KF↓ | HCA↑ | ARI↑ | NMI↑ | VM↑ |
| CIFAR100D | Threshold | 76.46±0.98 | 17.21±1.33 | 36.91±3.26 | 76.62±1.85 | **34.60±2.02** | 18.70±1.24 | 6.45±1.78 | 0.60±0.01 | 0.42±0.01 | 0.72±0.01 | 0.72±0.01 |
| | L2P (Wang et al., 2022b) | 59.93±2.15 | 8.57±2.49 | 43.10±4.30 | 50.53±7.25 | 9.60±1.50 | 27.39±2.11 | 27.85±7.21 | 0.50±0.02 | 0.37±0.01 | 0.69±0.05 | 0.69±0.05 |
| | DP (Wang et al., 2022a) | 66.09±1.01 | 8.80±1.69 | 48.34±6.78 | 56.19±2.00 | 8.68±2.06 | 28.93±2.25 | 29.06±1.99 | 0.55±0.01 | 0.44±0.01 | 0.72±0.00 | 0.72±0.00 |
| | GMP (Cendra et al., 2025) | 72.77±1.20 | 7.37±0.88 | 42.26±4.54 | 67.21±2.53 | 13.04±2.59 | 27.10±1.77 | 17.69±2.53 | 0.58±0.02 | 0.46±0.01 | 0.71±0.00 | 0.71±0.00 |
| | PHE (Zheng et al., 2024) | 72.35±1.12 | 16.74±1.05 | 37.34±1.31 | 70.12±1.07 | 14.55±1.01 | 24.65±1.35 | 2.12±0.92 | 0.58±0.01 | 0.42±0.00 | 0.70±0.01 | 0.70±0.01 |
| | Ours | **79.17±0.13** | **21.13±0.62** | **56.37±1.42** | **80.73±1.59** | 31.03±1.24 | **34.81±1.22** | 3.41±1.49 | **0.63±0.00** | **0.48±0.01** | **0.73±0.00** | **0.73±0.00** |
| CUB200D | Threshold | 66.09±1.20 | 43.60±2.08 | 44.05±4.96 | 65.52±3.68 | 49.90±1.33 | 6.64±0.67 | **1.61±3.55** | 0.48±0.00 | 0.20±0.00 | 0.70±0.00 | 0.70±0.00 |
| | L2P (Wang et al., 2022b) | 46.22±1.53 | 9.01±0.87 | 55.37±7.79 | 31.97±3.35 | 4.75±0.73 | 24.48±1.65 | 42.29±3.14 | 0.41±0.01 | 0.27±0.01 | **0.71±0.01** | **0.71±0.01** |
| | DP (Wang et al., 2022a) | 53.69±1.24 | 43.68±2.20 | 45.68±5.88 | 63.37±3.27 | 45.94±0.91 | 7.92±1.10 | 5.85±3.11 | 0.48±0.02 | 0.24±0.01 | 0.48±0.00 | 0.48±0.00 |
| | GMP (Cendra et al., 2025) | 62.97±1.33 | 46.44±1.87 | 47.99±4.34 | 58.11±3.00 | 48.02±1.20 | 10.31±1.45 | 5.46±2.77 | 0.48±0.01 | 0.26±0.01 | 0.71±0.00 | 0.71±0.00 |
| | PHE (Zheng et al., 2024) | 57.29±1.23 | 32.68±1.66 | **57.22±2.75** | 54.77±1.75 | 27.92±0.43 | 21.36±1.36 | 2.18±0.71 | 0.49±0.01 | 0.24±0.00 | 0.65±0.01 | 0.65±0.01 |
| | Ours | **66.20±0.55** | **58.30±2.37** | 43.33±6.10 | 64.42±0.65 | **65.28±1.78** | 37.25±4.90 | 4.07±0.47 | **0.50±0.01** | **0.27±0.01** | 0.70±0.00 | 0.70±0.00 |
| T-ImageNetD | Threshold | 57.53±1.80 | 11.35±1.56 | 63.31±3.55 | 52.36±3.10 | 6.48±1.40 | 13.81±2.11 | 22.90±3.08 | 0.24±0.01 | 0.15±0.01 | 0.34±0.00 | 0.34±0.00 |
| | L2P (Wang et al., 2022b) | 46.25±1.41 | 7.79±2.92 | 53.55±6.47 | 29.33±3.77 | 10.38±2.42 | 23.28±0.79 | 47.97±3.77 | 0.43±0.01 | 0.33±0.01 | 0.69±0.00 | 0.69±0.00 |
| | DP (Wang et al., 2022a) | 46.51±0.58 | 6.41±0.93 | 58.27±6.10 | 28.53±3.33 | 9.63±2.10 | 26.80±1.38 | 47.57±3.32 | 0.42±0.01 | 0.32±0.01 | 0.68±0.01 | 0.68±0.01 |
| | GMP (Cendra et al., 2025) | 62.47±1.40 | 6.25±1.72 | 58.02±4.29 | 63.95±2.04 | 15.64±2.63 | 26.31±2.33 | 16.86±2.04 | **0.56±0.01** | **0.43±0.01** | 0.72±0.01 | 0.72±0.01 |
| | PHE (Zheng et al., 2024) | 63.28±1.29 | 13.54±1.20 | 70.33±3.80 | 58.39±1.14 | 12.80±1.05 | 19.42±1.45 | 1.88±0.35 | 0.45±0.01 | 0.34±0.01 | 0.64±0.01 | 0.64±0.01 |
| | Ours | **75.31±1.31** | **16.04±0.76** | **73.81±2.67** | **74.94±2.20** | **16.23±1.24** | **37.43±1.30** | **1.15±2.18** | 0.56±0.02 | 0.40±0.00 | 0.72±0.00 | 0.72±0.00 |
| AircraftD | Threshold | 44.62±1.15 | 24.79±1.31 | 40.33±2.50 | 45.84±1.14 | 25.66±1.05 | 25.71±1.45 | 9.37±1.05 | 0.32±0.01 | 0.20±0.01 | 0.46±0.01 | 0.46±0.01 |
| | L2P (Wang et al., 2022b) | 43.21±1.08 | 20.11±1.20 | 35.52±1.93 | 40.17±2.34 | 21.00±1.21 | 27.21±1.33 | 15.04±2.10 | 0.30±0.00 | 0.16±0.00 | 0.43±0.01 | 0.43±0.01 |
| | DP (Wang et al., 2022a) | 40.18±1.10 | 21.88±1.23 | 34.80±1.79 | 42.53±1.46 | 20.22±0.96 | 28.15±1.74 | 12.68±1.74 | 0.31±0.01 | 0.18±0.00 | 0.44±0.00 | 0.44±0.00 |
| | GMP (Cendra et al., 2025) | 41.26±1.04 | 21.12±0.67 | 35.10±2.50 | 32.56±1.75 | 21.25±1.02 | 26.34±1.48 | 22.65±2.25 | 0.27±0.00 | 0.17±0.00 | 0.42±0.01 | 0.42±0.01 |
| | PHE (Zheng et al., 2024) | 45.82±1.02 | 22.56±1.05 | **41.72±2.13** | 47.28±2.88 | 25.09±1.76 | 24.38±1.67 | 7.93±1.13 | 0.31±0.00 | 0.19±0.01 | 0.44±0.01 | 0.44±0.00 |
| | Ours | **50.35±1.08** | **28.01±0.85** | 40.23±2.22 | **49.77±1.94** | **29.80±1.25** | 28.74±1.53 | 5.44±1.56 | **0.33±0.00** | **0.21±0.01** | **0.46±0.00** | **0.46±0.00** |

cluster space $\mathcal{Y}_{\text{seen}}$ does not directly align with the ground-truth label space $\mathcal{Y}_{\text{seen}}^{\text{GT}}$. To address this, we introduce two **agreement metrics**. Let $\mathcal{D}_c^{\text{test}}$ denote the subset of test samples with true label $c \in \mathcal{Y}_{\text{seen}}^{\text{GT}}$, and $\mathcal{C}_q^{\text{test}}$ the cluster assigned label $q \in \mathcal{Y}_{\text{seen}}$. With $p(x)$ as the predicted cluster of sample $x$, the metrics are: (1) *True-label Agreement ratio* (**TA**). This metric measures the maximum proportion of samples from a given true class that are predicted as the same class: $\text{TA} = \mathbb{E}_{c \in \mathcal{Y}_{\text{seen}}^{\text{GT}}} \frac{1}{|\mathcal{D}_c^{\text{test}}|} \max_{q \in \mathcal{Y}_{\text{seen}}} (\sum_{x \in \mathcal{D}_c^{\text{test}}} \mathbf{1}(p(x) = q))$. (2) *Cluster Agreement ratio* (**CA**). This metric measures the maximum proportion of samples from a given predicted cluster that are with the same true label: $\text{CA} = \mathbb{E}_{q \in \mathcal{Y}_{\text{seen}}} \frac{1}{|\mathcal{C}_q^{\text{test}}|} \max_{c \in \mathcal{Y}_{\text{seen}}^{\text{GT}}} (\sum_{(x,y) \in \mathcal{C}_q^{\text{test}}} \mathbf{1}(y = c))$. Together, TA and CA capture complementary aspects of novel-class discovery, and a method that achieves jointly higher trade-off demonstrates a more reliable and effective discovery. For completeness, we also provide *post-hoc evaluation* following NCD protocols, including Hungarian Cluster Accuracy (HCA) (Meilă, 2003), Adjusted Rand Index (ARI) (Rand, 1971), Normalized Mutual Information (NMI) (McDaid et al., 2011) and V-Measure (Rosenberg & Hirschberg, 2007). See more in Appendix. C.

## 5.2 MAJOR COMPARISONS

In this paper, we first compare our methods with some thresholding-based training-free methods using the Euclidean distance between prototypes We also compare with some training-required methods, including L2P (Wang et al., 2022b), DP (Wang et al., 2022a), and GMP (Cendra et al., 2025) and PHE (Zheng et al., 2024), these methods update prompts like TTA and CL methods. The results are shown in Table 3, and we have some major observations. First, real-time and post evaluations exhibit notable differences, with post evaluations showing higher TA but lower CA. This suggests that in NCD with large sample sizes, post evaluation better distinguishes class differences, grouping similar samples into clusters. However, clusters often contain multiple classes, indicating confusion in new class discovery. Second, compared to traditional threshold-based training-free methods, training-based methods suffer from performance degradation. Immediate model updates upon discovering new classes lead to lower TA and CA, along with more severe catastrophic forgetting. Third, our method outperforms others across all three datasets, demonstrating that by leveraging fine-grained sample-level comparisons and a memory self-correction mechanism, our approach enhances adaptability in novel class discovery. Finally, the traditional NCD metrics indicate that the proposed method still performs well when evaluated using the classic NCD clustering accuracy metric.

## 5.3 ANALYSIS ON TTD

**Different number of discoverable categories.** In our experiments, we set an upper limit on discoverable classes, though real-world scenarios may involve a much larger or even infinite number. We tested three cases: (1) reducing unknown classes while keeping known classes fixed, (2) increasing known classes while keeping unknown classes unchanged, and (3) exceeding the true number of unknown classes. Table 4 presents the results. We find that more discoverable classes generally improve TA and CA, though the effect depends on the number of known classes. When the discoverable

Table 4: Comparisons of different known (K) and unknown class numbers (U).

| K+U | Real-time Eval | | Post Eval | | |
|---|---|---|---|---|---|
| | TA | CA | TA | CA | KF |
| 70+30 | 21.11 | 56.87 | 31.03 | 34.81 | 3.47 |
| 80+20 | 11.57 | 64.15 | 20.90 | 31.09 | 0.69 |
| 90+10 | 14.92 | 52.98 | 21.50 | 30.61 | 0.44 |
| 70+100 | 17.58 | 81.28 | 25.03 | 40.74 | 6.46 |
| 70+200 | 19.86 | 85.60 | 26.87 | 42.63 | 7.79 |
| 70+∞ | 20.37 | 92.94 | 22.63 | 47.09 | 10.69 |

Table 5: HM performance under extreme and imbalanced test-time distributions.

| Distrubution | Real-time Eval | | Post Eval | | |
|---|---|---|---|---|---|
| | TA | CA | TA | CA | KF |
| All classes long-tail | 18.77 | 52.26 | 25.13 | 31.20 | 7.76 |
| Unknown long-tail | 24.69 | 54.13 | 27.51 | 34.37 | 5.55 |
| Unknown at begining | 24.69 | 59.13 | 32.78 | 36.49 | 4.68 |
| Unknown at end | 19.23 | 53.41 | 22.35 | 32.26 | 6.82 |
| Randomly (Ours) | 21.11 | 56.87 | 31.03 | 34.81 | 3.47 |

classes far exceed the true number, real-time evaluation shows TA and CA gains, but post-evaluation sees a TA drop and more severe forgetting. These findings highlight the importance of appropriately setting the number of discoverable classes for optimal TTD performance.

**Robustness under extreme distributions**. Table 5 reports stress-test results under long-tail and staged-arrival settings. Performance drops most in the all-class long-tail case (TA 18.8%), confirming the difficulty of rare-class discovery. In contrast, when unknowns arrive at the beginning or end, HM maintains relatively stable TA/CA, and the random order yields the best overall balance. KF remains consistently low, showing that known-class recognition is robust across scenarios.

**Human annotation vs. Auto label assignment**. In the above experiments, newly discovered classes are automatically assigned non-semantic labels. When human annotations are provided for samples potentially belonging to new classes, semantic labels eliminate erroneous discoveries. Table 6(a) presents this experiment, showing that with human annotations, the TA for novel class discovery significantly improved (21.11 vs. 52.10), while CA declined. This suggests that human annotations strongly enhance intra-class consistency but do not improve real-time classification performance.

**Training-free vs. training-required TTD**. In Table 6(b), we explore the impact of allowing prompt training while using HM. The results reveal a sharp decline in CA and a rise in TA after training, suggesting that most new classes are misclassified as known ones. This highlights how model updates in TTD directly affect known class performance, leading to a rapid degradation in overall recognition when new classes remain uncertain.

## 5.4 ANALYSIS ON HASH MEMORY AND LSH CLASSIFIER

**Trends of TA and CA**. In Fig. 4, we show a continuous decline in TA and a steady increase in CA, aligning with the TTD process. Early on, clusters contain fewer samples, leading to more pseudo-labeling errors. Over time, as more test samples accumulate, TA and CA gradually reach a balance.

**PCA direction bases vs. random direction bases**. In our method, we use randomly assigned reference directions when constructing hash values. In Table 6(c), we test an alternative approach using PCA to obtain reference directions from known classes. The results show that PCA-based directions perform similarly to random directions.

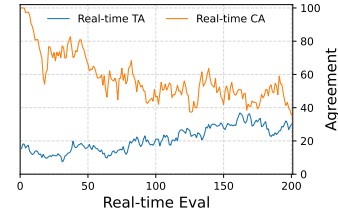

Figure 4: Trends of TA and CA.

**Different memory sizes**. Table 7 presents the effect of different memory sizes in HM. As memory size increases, TA decreases while CA increases, suggesting that excessive memory may impair the correct rediscovery of new classes. This is likely due to the reliance on pseudo-labeling in memory, where rapidly storing a large number of new class samples increases the risk of misclassification.

**Ablation study**. Table 8 reports ablations based on a baseline that uses a confidence threshold and EMA-updated class prototypes. Adding HM yields notable gains, especially in CA, by improving the consistency of discovered clusters. Incorporating SC provides further improvement by correcting early misassignments and increasing the reliability of stored samples.

**Memory agreement w/ and w/o SC**. In Fig. 5, we compare the impact of using and not using the SC strategy on memory. The left plot shows the number of true novel class samples in the buffer, while the right plot illustrates the consistency of true labels within each buffer class. The results indicate

Table 6: Three diverse settings.

| Experiment | Real-time Eval | | Post Eval | | |
|---|---|---|---|---|---|
| | TA | CA | TA | CA | KF |
| (a) Human | 52.10 | 42.96 | 48.27 | 49.44 | 5.81 |
| (b) Train | 43.25 | 12.88 | 9.83 | 33.69 | 3.17 |
| (c) PCA | 20.86 | 46.33 | 28.25 | 28.91 | 3.24 |
| Ours | 21.11 | 56.87 | 31.03 | 34.81 | 3.47 |

Table 7: Different memory size.

| Memory size (per class) | Real-time Eval | | Post Eval | | |
|---|---|---|---|---|---|
| | TA | CA | TA | CA | KF |
| 0 | 26.56 | 40.85 | 30.60 | 29.42 | 5.43 |
| 10 | 22.33 | 53.18 | 28.47 | 28.02 | 3.71 |
| 20 | 21.11 | 56.87 | 31.03 | 34.81 | 3.47 |
| 30 | 15.39 | 66.99 | 22.57 | 33.05 | 3.44 |

Table 8: Ablation study.

| HM | SC | Real-time Eval | | Post Eval | | |
|---|---|---|---|---|---|---|
| | | TA | CA | TA | CA | KF |
| | | 26.56 | 40.85 | 30.60 | 29.42 | 5.43 |
| ✓ | | 22.18 | 52.51 | 31.21 | 31.01 | 4.79 |
| ✓ | ✓ | 21.11 | 56.87 | 31.03 | 34.81 | 3.47 |

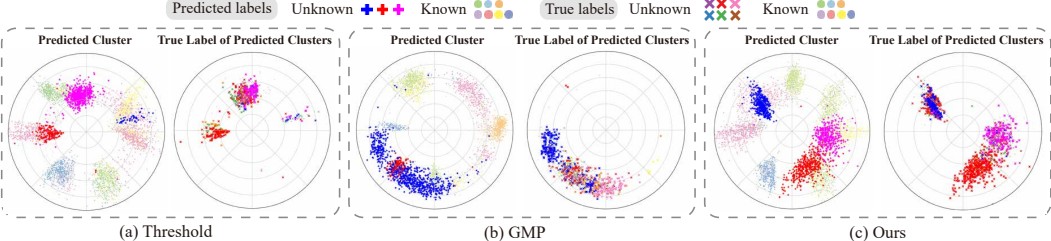

Figure 5: Memory agreement w/ and w/o SC.   Figure 6: Frequency of memory self-correction.

that SC effectively corrects mislabeled samples in the buffer, enhancing memory performance.

**Frequency of SC**. In Fig. 6, we examine the effect of different SC frequencies. The results show that frequent self-correction increases computation time without significantly improving novel class discovery. Conversely, infrequent self-correction leads to the accumulation of erroneous pseudo-labels, negatively impacting model performance.

**Post visualization using t-SNE**. In Fig. 7, we employ t-SNE (Van der Maaten & Hinton, 2008) to visualize the true-label distribution of test samples, comparing predicted clusters with their corresponding truth labels. The results demonstrate that our method effectively distinguishes between known and unknown classes, whereas baseline methods often confuse them, compromising both novel class discovery and known class classification. Additionally, our approach successfully groups samples with the same labels into cohesive clusters, a crucial factor for effective novel class discovery.

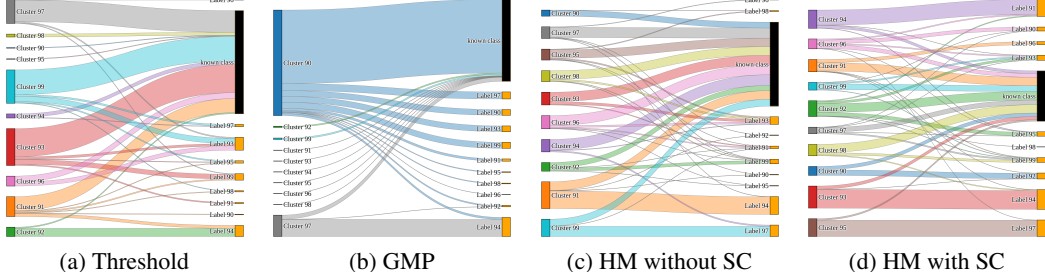

Figure 7: Post t-SNE visualization in angular space on CIFAR100D with 7 known and 3 unknown classes. The first column is colored by predicted clusters. In the second column, within each cluster, the true label that appears most frequently is assigned the same color as the cluster in the first column.

**Matching between true-label and cluster.** In Fig. 8, we show the label-cluster of unknown classes. We find that using thresholding methods and GMP are easy to predict samples with unknown class as known classes. HM reduces this phenomenon and can make better discoveries with the SC module.

Figure 8: Comparison of unknown class label matching between predictions and truth labels across different methods (Cifar-100D 90+10). The right black bars represent misclassified old class samples.

## 6 CONCLUSION

This paper introduced TTD, a real-time evaluation protocol for novel class discovery that complements the post-hoc evaluation of NCD and the distribution-shift focus of test-time training. To address the unique challenges of TTD, we proposed a training-free hash memory method that combines semantic-aware hashing, collaborative global-to-local classification, and a lightweight self-correction mechanism. Extensive experiments on diverse benchmarks demonstrated that HM achieves more accurate and stable real-time discovery while maintaining performance on known classes, establishing a strong baseline under the TTD protocol. Despite these gains, our method relies on dynamically maintaining a memory buffer during testing, which may be unsuitable in privacy-sensitive applications.

ETHICS STATEMENT

This work does not involve human subjects, sensitive data, or other aspects that may raise ethical concerns. Therefore, we believe there are no ethical issues associated with our study.

REPRODUCIBILITY STATEMENT

We provide the source code in the supplementary zip to ensure reproducibility. The paper includes detailed descriptions of dataset construction, algorithmic designs, and experimental procedures, allowing others to reproduce and verify our results.

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

# Appendix

## A  DETAILED RELATED WORK

### A.1  TEST-TIME LEARNING

*Test-Time Training* (TTT) is a machine learning approach where a model continues to update its parameters during the test phase by leveraging auxiliary tasks. These tasks, often self-supervised, are designed to exploit the structure of the input data and improve the model's adaptability to test-time distribution shifts. By incorporating additional training during inference, TTT can adjust specific components, such as feature extractors, to enhance performance on out-of-distribution or domain-shifted data. TTT (Sun et al., 2020) trains a model jointly on the task loss and an auxiliary image-rotation prediction task during training, which couples the gradients of the auxiliary task and the task itself. TTT++ (Liu et al., 2021) further enhances performance during test time. The statistics of the source data are calculated from a retained queue of source feature maps. Subsequently, these statistics are employs to align with the target data, thereby regularizing the contrastive loss. TTT-MAE (Gandelsman et al., 2022) utilized a masked Autoencoder to predict the obscured parts, thus tuning the model at test time. TTTFlow (Osowiechi et al., 2023) employs Normalizing Flows to detect domain shifts and adapt deep neural networks for improved accuracy on distributionally shifted data. TTAC++ (Su et al., 2024) employs anchored clustering to discover clusters in both source and target domains and matches the target clusters to the source ones for improved adaptation. When source domain information is strictly absent, the method can efficiently infer source domain distributions to achieve source-free test-time training. However, this approach typically requires additional computation and well-designed auxiliary objectives.

*Test-Time Adaptation* (TTA) focuses on dynamically adjusting a model during inference without explicit training. This method usually updates lightweight parameters, such as batch normalization statistics, or applies simple strategies like entropy minimization to adapt the model to new test data distributions. TENT (Wang et al., 2020) conducts its adaptation on the batch normalization layers using the conditional entropy loss of the predictions. SHOT (Liang et al., 2020) only needs a well-trained source model and achieves domain adaptation without accessing potentially private source data by leveraging information maximization and self-supervised pseudo-labeling techniques. LAME (Boudiaf et al., 2022) adjusts the classifier's output rather than its internal parameters, utilizing Laplacian regularization to encourage consistent latent assignments for neighboring points in the feature space. AdaContrast (Chen et al., 2022) refines pseudo labels through soft voting among nearest neighbors in the target feature space and utilizes contrastive learning to exploit pairwise information among target samples. RoTTA (Yuan et al., 2023) employs a robust batch normalization scheme to estimate normalization statistics and utilizes a memory bank to capture a snapshot of the test distribution through category-balanced sampling that considers timeliness and uncertainty. Moreover, RoTTA develops a timeliness-aware reweighting strategy along with a teacher-student model to stabilize the training process. TTA aims to improve the model's robustness and accuracy while keeping computational overhead minimal, making it particularly suitable for real-time applications or scenarios with mild distribution shifts.

In summary, TTT and TTA tasks focus solely on handling changes in data distribution and do not consider scenarios where new classes are encountered during testing. In open-world testing environments, models are likely to encounter new classes at any time, and the ability to autonomously discover these new classes would enhance the model's adaptability to a broader range of unknowns.

### A.2  NOVEL CATEGORY DISCOVERY

Novel Category Discovery (NCD) focuses on enabling models to autonomously detect and adapt to new, previously unseen classes during the inference phase, particularly in open-world environments. NCD allows models to identify when a new category appears and respond appropriately, often without labeled data for the new class. The challenge lies in ensuring that models can effectively detect and represent novel categories while maintaining the integrity of previously learned knowledge. DTC (Han et al., 2019) leverages prior knowledge of related but distinct image classes, reduces ambiguity in clustering, and enhances the quality of newly discovered classes. Han et al. (Han et al., 2020) introduced a novel approach that combines self-supervised learning, ranking statistics, and joint

optimization to automatically discover and learn new visual categories in an image collection. Zhao et al. (Zhao & Han, 2021) proposed using dual ranking statistics and mutual knowledge distillation, generating pseudo labels through a dynamically constructed local part dictionary, and allowing information exchange between the two branches to encourage agreement on new category discovery, thus enabling the model to leverage the benefits of both global and local features. GCD (Vaze et al., 2022) leverages contrastively trained Vision Transformers to directly assign labels through clustering, avoiding overfitting to labeled classes. PromptCAL (Zhang et al., 2023) performs contrastive affinity learning with auxiliary prompts and attribute semantic self-supervised learning to enhance visual feature discrimination for generalized novel category discovery. PromptCCD (Cendra et al., 2025) employs a Gaussian Mixture Model (GMM) as a prompt pool, utilizing a dynamic Gaussian Mixture Prompting (GMP) module to facilitate representation learning and mitigate forgetting.

Unlike traditional classification tasks, NCD does not require immediate labeling of new classes. Instead, *it uses clustering on additional test data for post-hoc evaluation of new class discovery*. This makes current NCD methods *unsuitable* for test-time application, as they rely on further clustering analysis, whereas test-time requires immediate results and the ability to incorporate those results into further training without a separate evaluation phase.

## B DATASET CONSTRUCTION

We conduct our experiments using three widely recognized benchmark datasets: CIFAR100 (C100) (Krizhevsky et al., 2009), Caltech-UCSD Birds-200-2011 (CUB) (Wah et al., 2011), Tiny ImageNet (Le & Yang, 2015) and FGVC-Aircraft (Maji et al., 2013). Each of these datasets is systematically partitioned into known and unknown classes. The model undergoes training on the known training set and is subsequently evaluated on a mixed set containing both known and unknown classes. Since the primary objective of these datasets is to facilitate new class discovery, we denote the transformed versions as CIFAR100D, CUB-200D, and Tiny-ImageNetD to reflect this adaptation.

Table 9: Statistic of the used datasets (K: Known, U: Unknown). We set batch size to 50 for all datasets.

| Dataset | Labeled | CIFAR100D | | | CUB-200D | | | Tiny-ImagenetD | | | AircraftD | | |
|---------|---------|---|---|---|---|---|---|---|---|---|---|---|---|
| | | K | U | #sample (stream length) | K | U | #sample (stream length) | K | U | #sample (stream length) | K | U | #sample (stream length) |
| TrainSet | ✓ | 70 | 0 | 35000 | 140 | 0 | 4195 | 140 | 0 | 70000 | 70 | 0 | 2331 |
| TestSet | | 70 | 30 | 10000 (200 batches) | 140 | 60 | 5794 (115 batches) | 140 | 60 | 10000 (200 batches) | 70 | 30 | 3333 (66 batches) |

The dataset partitioning follows the scheme outlined in Table 9. Specifically, during the training phase, we divide the training set into known and unknown classes based on their class index order. To ensure stable behavior in the early stage of the test stream when only few unknown samples have been observed, we adopt a 7:3 known to unknown ratio as the primary configuration, which provides sufficient known class structure for initialization while retaining a substantial portion of unseen categories for online discovery. For instance, in CIFAR100, the first 70 classes are designated as known, while the remaining 30 classes are treated as unknown. The supervised training process is then conducted using only the known classes within the labeled training set. More precisely, CIFAR100D consists of classes 0–69 (70 known classes in total), CUB-200D includes classes 0–139 (140 known classes in total), Tiny-ImageNetD comprises classes 0–139 (140 known classes in total), and AircraftD includes classes 0-69 (70 known classes in total), all of which are utilized for training.

During the test phase, the model is evaluated on the entire unlabeled test set, which includes samples from all categories, enabling new class discovery and classification. While the category labels remain structured according to the original known-unknown splits (e.g., 70+30 for CIFAR100D and AircraftD and 140+60 for CUB-200D and Tiny-ImageNetD), these labels are only used for metric evaluation and are not provided to the model during inference. This setup ensures a realistic scenario for open-world learning, where the model must autonomously identify and categorize previously unseen classes.

# C  METRIC DEFINITION

The evaluation process consists of two parts: one for known classes and one for unknown classes. In the TTD setting, real-time evaluation is essential, so all metrics are computed incrementally as the model processes each test sample. This provides an online view of classification quality and discovery behavior throughout the stream, and we report both the evolving curves and the final accumulated scores over the full test sequence. Classical NCD, GCD, and OCD metrics cannot be directly applied in this setting because they require access to the entire test set and are therefore only definable post-hoc evaluation. To enable evaluation under streaming constraints, we introduce agreement-based metrics that capture the same notions of cluster–label consistency without violating the test-time protocol. For completeness, we also include post-evaluation results following standard NCD practice, where all predictions are revalidated once the entire test set has been processed. This post-hoc evaluation leverages the full data distribution and is provided solely for comparability with prior discovery frameworks.

## C.1  METRICS FOR KNOWN CLASSES

For the evaluation of known classes, we employ two key metrics to comprehensively assess the model's performance: Known Accuracy (KA) and Known Forgetting (KF).

(1) *Known Accuracy* (**KA**). KA measures the traditional classification accuracy of the model on known classes, reflecting its ability to correctly recognize and classify samples that were part of the training set. This metric serves as a standard benchmark for evaluating the retention of previously learned knowledge:

$$\text{KA} = \mathbb{E}_{c \in \mathcal{Y}_{\text{known}}} \frac{1}{|\mathcal{D}_c^{\text{test}}|} \sum_{x \in \mathcal{D}_c^{\text{test}}} \mathbf{1}(p(x) = c), \tag{7}$$

where $\mathcal{Y}_{\text{known}}$ is set of predefined known classes, $\mathcal{D}_c^{\text{test}}$ is test samples with ground-truth class $c$, $p(x)$ is the predicted label for sample $x$, $\mathbf{1}(\cdot)$ is the indicator function (1 if prediction matches true class $c$, 0 otherwise)

(2) *Known Forgetting* (**KF**). KF, on the other hand, quantifies the degree of performance degradation on known classes over time. It captures the extent to which the model forgets previously learned information as it encounters new data, particularly when adapting to novel classes. A lower KF score indicates better knowledge retention, while a higher score suggests significant forgetting.

$$\text{KF} = \text{KA}_{\text{post}} - \text{KA}_{\text{pre}}, \tag{8}$$

where $\text{KA}_{\text{post}}$ and $\text{KA}_{\text{pre}}$ are the KA computed on all test data with known classes, before and after TTD.

## C.2  METRICS FOR UNKNOWN CLASSES

For unknown classes, since the predicted label space $\mathcal{Y}_{\text{seen}}^{\text{GT}}$ does not match the cluster label space $\mathcal{Y}_{\text{seen}}$, we propose agreement metrics to assess effectiveness. In the test set $\mathcal{D}^{\text{test}}$, a sample $x$ has a true label $y \in \mathcal{Y}_{\text{seen}}^{\text{GT}}$ and a predicted cluster label $p(x) \in \mathcal{Y}_{\text{seen}}$. We define the subset of $\mathcal{D}^{\text{test}}$ with true label $c$ as $\mathcal{D}_c^{\text{test}}$, and the cluster with predicted label $p$ as $\mathcal{C}_p^{\text{test}}$.

(1) *True-label Agreement ratio* (**TA**). This metric measures the maximum proportion of samples from a given true class that are predicted as the same class:

$$\text{TA} = \mathbb{E}_{c \in \mathcal{Y}_{\text{seen}}^{\text{GT}}} \frac{1}{|\mathcal{D}_c^{\text{test}}|} \max_{p \in \mathcal{Y}_{\text{seen}}} \left( \sum_{x \in \mathcal{D}_c^{\text{test}}} \mathbf{1}(p(x) = p) \right), \tag{9}$$

where $\mathbf{1}(\cdot)$ is the indicator function (1 if true, 0 otherwise).

(2) *Cluster Agreement ratio* (**CA**). This metric measures the maximum proportion of samples from a given predicted cluster that are with the same true label:

$$\text{CA} = \mathbb{E}_{p \in \mathcal{Y}_{\text{seen}}} \frac{1}{|\mathcal{C}_p^{\text{test}}|} \max_{c \in \mathcal{Y}_{\text{seen}}^{\text{GT}}} \left( \sum_{(x,y) \in \mathcal{C}_p^{\text{test}}} \mathbf{1}(y = c) \right). \tag{10}$$

### C.3 NCD METRICS

Traditional novel class discovery (NCD) methods typically rely on post-cluster evaluation, where the quality of the discovered clusters is assessed after the entire test set has been processed. To ensure a comprehensive comparison with existing approaches, we also report several widely used clustering evaluation metrics, including Hungarian Cluster Accuracy (HCA) (Meilă, 2003), Adjusted Rand Index (ARI) (Rand, 1971), Normalized Mutual Information (NMI) (McDaid et al., 2011), and V-Measure (Rosenberg & Hirschberg, 2007). Note that these metrics are only evaluated after TTD, say post evaluation.

(1) *Hungarian Cluster Accuracy* (**HCA**). This metric measures the clustering accuracy by computing an optimal one-to-one mapping between predicted clusters and ground-truth labels using the Hungarian algorithm. It provides an intuitive evaluation of how well the discovered clusters align with the actual class distributions. HCA can be computed as

$$\text{HCA} = \mathbb{E}_{(x,y)\in\mathcal{D}^{\text{test}}}(y = \text{map}(p(x))), \tag{11}$$

where $\text{map}(\cdot)$ is the optimal mapping from clustering to true labels obtained based on the Hungarian algorithm

(2) *Adjusted Rand Index* (**ARI**). ARI quantifies the similarity between the predicted clustering assignments and the ground-truth labels while adjusting for chance. It accounts for both correct pairwise clustering and misclustered pairs, offering a robust measure of clustering consistency.

$$\text{ARI} = \frac{\text{RI} - \mathbb{E}[\text{RI}]}{\max(\text{RI}) - \mathbb{E}[\text{RI}]}, \tag{12}$$

where Rand Index $(\text{RI}) = \frac{a+b}{C_n^2}$, $a$ is the logarithm of samples of the same class assigned to the same cluster, and $b$ is the logarithm of samples of different classes assigned to different clusters. $n$ is the total number of samples, combination $C_n^2 = \frac{n(n-1)}{2}$ and $\mathbb{E}[\text{RI}]$ is the expected value of RI.

(3) *Normalized Mutual Information* (**NMI**). NMI assesses the mutual dependence between predicted and true labels by measuring the shared information between the two distributions. A higher NMI value indicates better alignment between the discovered clusters and the actual categories. The value interval of NMI is [0,1], and a larger value indicates a higher degree of information sharing between the clustering results and the real labels.

$$\text{NMI}(\mathcal{U},\mathcal{V}) = \frac{2 \cdot I(\mathcal{U};\mathcal{V})}{H(\mathcal{U}) + H(\mathcal{V})}, \tag{13}$$

where $\mathcal{U}$ is collection of true labels and $\mathcal{V}$ is collection of predictions. $I(\mathcal{U};\mathcal{V})$ is Mutual Information where $I(\mathcal{U};\mathcal{V}) = H(\mathcal{U}) - H(\mathcal{U}|\mathcal{V})$. $H(\mathcal{U})$ is the entropy of true label, $H(\mathcal{U}) = -\sum_{c=1}^{C} p(c)\log p(c)$, and $H(\mathcal{V})$ is the entropy of prediction, $H(\mathcal{V}) = -\sum_{k=1}^{K} p(k)\log p(k)$.

(4) *V-Measure* (**VM**). The VM is taken in the interval [0,1], which simultaneously constrains the purity and coverage of the clusters through the harmonic mean. Both VM and NMI are symmetric metrics that support the comparison of clusters and categories at different scales.

$$\text{V-Measure} = \frac{2 \cdot h \cdot c}{h + c}, \tag{14}$$

where homogeneity $h = 1 - \frac{H(\mathcal{U}|\mathcal{V})}{H(\mathcal{U})}$, and completeness $c = 1 - \frac{H(\mathcal{V}|\mathcal{U})}{H(\mathcal{V})}$. $H(\mathcal{U}|\mathcal{V}) = -\sum_{k=1}^{K}\sum_{t=1}^{T} p(k,t)\log\frac{p(k,t)}{p(k)}$ and $H(\mathcal{V}|\mathcal{U}) = -\sum_{t=1}^{T}\sum_{k=1}^{K} p(t,k)\log\frac{p(t,k)}{p(t)}$, where $p(t) = \frac{N_t}{N}$ is the sample proportion of class $t$, $p(k) = \frac{N_k}{N}$ is the sample proportion of cluster $k$, and $p(t,k) = \frac{count_k(t)}{N}$ is the joint distribution probability.

## D HYPER-PARAMETER ANALYSIS

### D.1 NUMBER OF SELECTED NEIGHBORING BUCKETS IN GRAPH-BASED NEIGHBOR SEARCHING

In Fig. 9, we present a comparative analysis of the impact of varying the number of neighbor buckets in our method during TTD. The choice of bucket size directly influences both the effectiveness of

new class discovery and computational efficiency. When the number of buckets is small, the model has access to a limited number of nearest neighbors. This scarcity of reference samples hinders the model's ability to effectively discover and categorize novel classes, as the available nearest-neighbor information may be insufficient to form meaningful clusters. However, a smaller bucket size also results in lower computational overhead, as fewer comparisons are required during the clustering process. Conversely, when the number of buckets is large, the model can consider a greater number of nearest neighbors. While this increases the availability of reference samples for new class discovery, an excessive number of neighbors can introduce noisy or misleading information, leading to incorrect clustering assignments. Additionally, the increased volume of sample comparisons significantly raises computational costs, prolonging processing time. Thus, selecting an optimal number of neighbor buckets is a trade-off between discovery accuracy and computational efficiency. A balanced choice ensures that the model captures sufficient nearest-neighbor information while minimizing erroneous interferences and excessive time consumption.

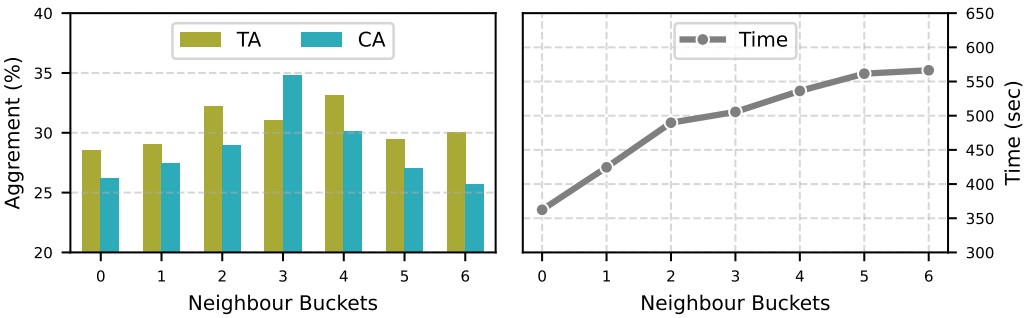

Figure 9: Comparisons of different numbers of selected neighboring buckets.

### D.2 NUMBER OF COMPARED SAMPLES IN BUCKETS

After retrieving the associated bucket using the hash value, we calculate the distances between the samples within the bucket and the target test sample. The top-$k$ nearest samples are then identified, and their labels are used for voting to determine the final classification of the target test sample. Thus, selecting an appropriate number of voting samples, i.e., determining the optimal value of $k$, is a critical factor in achieving accurate classification. To investigate the impact of different $k$ values, we conduct comparative experiments with varying settings, as presented in Table 10. The results reveal that when $k$ is relatively small, TA decreases, whereas CA improves. This suggests that samples from the same class are more likely to be concentrated within known classes, while the number of correctly captured samples in new class clusters remains limited. On the other hand, as $k$ increases, both CA and TA decline, indicating that the internal structure of new class clusters becomes more confused. This implies that an excessively large $k$ introduces more noise, making it harder for the model to differentiate between novel categories. These findings underscore the importance of carefully selecting an appropriate $k$ value to maintain effective discovery.

Table 10: Different number of compared samples in buckets on CIFAR100D.

|  | Real-time Eval | | Post Eval | | |
|---|---|---|---|---|---|
|  | TA | CA | TA | CA | KF |
| 0 | 19.82 | 39.27 | 22.76 | 25.80 | 6.22 |
| 5 | 18.05 | 56.06 | 35.26 | 27.83 | 7.10 |
| 10 | 21.11 | 56.87 | 31.03 | 34.81 | 3.47 |
| 15 | 16.73 | 52.15 | 30.40 | 28.67 | 4.43 |
| 20 | 13.79 | 54.24 | 29.23 | 26.15 | 6.32 |

### D.3 NUMBER OF RANDOM DIRECTIONS WHEN HASHING FEATURES

When constructing the hash memory, we utilize multiple random directional bases to define feature orientations and partition the angular space. The number of these bases plays a crucial role in performance. To investigate its impact, we conduct comparative experiments with different base quantities, as shown in Fig. 10. The results indicate that when the number of bases is too low, the angular space is insufficiently separated, leading to decreased TA and CA. Additionally, the reduced ability to distinguish between different feature directions increases comparison time, further affecting efficiency. On the other hand, when the number of bases is too high, we observe a similar decline in performance. This suggests that an excessive number of buckets does not improve sample-level discrimination. Instead, it increases the complexity of searching within nearest-neighbor buckets, making comparisons less effective and computationally more demanding. These findings emphasize the importance of choosing an optimal number of random directional bases to achieve a balanced trade-off between discovery effectiveness, and computational efficiency.

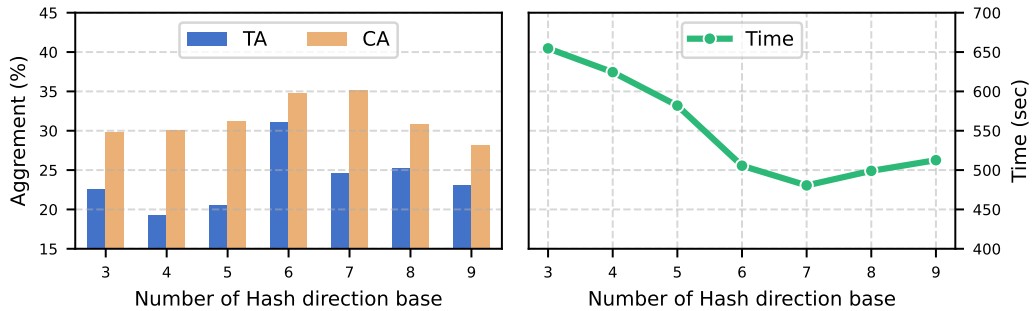

Figure 10: Metrics of different Hash directions.

### D.4 FACTOR FOR EXPONENTIAL MOVING AVERAGE (EMA)

Although our method does not update the model itself, it continuously refines the prototype representations of each class. To assess the impact of prototype updates on both known and seen classes, we conduct experiments, with the results presented in Tables 11 and 12. The findings indicate that an overly aggressive update of newly discovered class prototypes amplifies errors introduced during the discovery process, leading to a decline in TA. For known classes, the most effective approach is to maintain static prototypes without updates. Updating these prototypes not only exacerbates catastrophic forgetting but also increases confusion between known and unknown classes, further deteriorating overall performance.

Table 11: Comparisons of different EMA factors for unknown classes.

| $\alpha$ | Real-time Eval | | Post Eval | | |
|---|---|---|---|---|---|
| | TA | CA | TA | CA | KF |
| 1.0 | 14.55 | 65.83 | 5.17 | 44.31 | 0.71 |
| 0.99 | 14.57 | 66.91 | 22.80 | 36.92 | 2.59 |
| 0.9 | 21.11 | 56.87 | 31.03 | 34.81 | 3.47 |
| 0.8 | 21.07 | 55.75 | 25.30 | 31.57 | 3.26 |
| 0.7 | 19.23 | 57.19 | 16.27 | 29.18 | 2.43 |
| 0.6 | 19.74 | 56.43 | 16.97 | 27.42 | 2.74 |
| 0.5 | 19.68 | 58.83 | 13.27 | 32.16 | 2.00 |

### D.5 DISCRETIZATION COEFFICIENT

In Eq. (1), we use a discretization coefficient $\kappa$ to quantizes the feature norm into bins ($\lfloor \kappa \|f(x)\| \rfloor$), ensuring a compact hash space. This design significantly improves clustering stability in the memory.

Table 12: Comparisons of different EMA factors for known classes.

| $\alpha$ | Real-time Eval | | Post Eval | | |
|---|---|---|---|---|---|
| | TA | CA | TA | CA | KF |
| 1.0 | 21.11 | 56.87 | 31.03 | 34.81 | 3.47 |
| 0.99 | 16.55 | 60.05 | 19.13 | 30.60 | 3.47 |
| 0.9 | 14.54 | 59.65 | 10.53 | 32.06 | 2.76 |
| 0.8 | 16.19 | 55.21 | 14.77 | 32.50 | 4.71 |
| 0.7 | 19.51 | 47.82 | 17.07 | 29.33 | 16.49 |
| 0.6 | 17.93 | 47.34 | 19.83 | 25.69 | 22.33 |
| 0.5 | 19.04 | 48.74 | 19.90 | 20.99 | 25.69 |

In Table 13, we show the impact of different discretization coefficient. The results show that large $\kappa$ over-amplifies norm differences, harming known-class accuracy and overall stability. Small $\kappa$ reduces cluster alignment. A moderate $\kappa$ offers the best trade-off between known-class preservation and novel-class discovery.

Table 13: Discretization coefficient $\kappa$ selection.

| $\kappa$ | Real-time Eval | | | Post Eval | | | |
|---|---|---|---|---|---|---|---|
| | KA | TA | CA | KA | TA | CA | KF |
| 1 | 80.22 | 23.02 | 54.18 | 81.02 | 32.29 | 31.24 | 3.22 |
| 2 | 79.20 | 21.11 | 56.87 | 80.77 | 31.03 | 34.81 | 3.47 |
| 4 | 76.52 | 20.45 | 55.16 | 78.28 | 18.21 | 29.78 | 5.96 |
| 10 | 77.56 | 25.41 | 50.84 | 75.25 | 22.32 | 24.65 | 8.99 |

# E ANALYSIS ON LSH CLASSIFIER

## E.1 OTHER DESIGNS OF HASH CODING

In the TTD setting, the model must continuously discover novel categories in a streaming, unlabeled environment. This makes the design of the hash code, i.e., how we map each feature to memory. Under such constraints, we found that existing memory and pseudo-labeling strategies were either unstable or computationally inefficient. Therefore, we designed our LSH-based approach guided by the following key task-level observations: We observe that samples from novel categories typically exhibit lower feature norms than known-class samples. To exploit this property, we use norm-based bits to encourage rough separation in the hash space. Empirical results in the following table confirm this norm gap and its utility. Moreover, norm separation alone is insufficient for discriminating between emerging novel classes. Therefore, we introduce direction-sensitive bits based on inner products with random anchors, and amplify norm-based differences with a tunable parameter $\kappa$. This design is meant to serve as a strong, low-cost benchmark that embodies core principles necessary for real-time novel category adaptation. As demonstrated in Table 14, it consistently outperforms other encoding strategies such as PCA, norm-only, direction-only, and random hashing. Thus, our method is a task-driven design that reflects the key dynamics of TTD, i.e., real-time decision making, representation uncertainty, and memory-guided self-organization.

## E.2 BOUNDARY BETWEEN PROTOTYPE AND LSH CLASSIFIERS

Our method employs a global-to-local classification approach, where confidence determines whether to use the prototype classifier or the LSH-based classifier. In Fig. 11, we analyze the effect of varying this boundary value. The results show that with a higher boundary, *i.e.*favoring the LSH-based classifier, KA drops sharply. Conversely, with a lower boundary, *i.e.*favoring the prototype classifier, novel class discovery becomes unstable.

Table 14: Other design of hash coding.

| | Real-time Eval | | | Post Eval | | | |
|---|---|---|---|---|---|---|---|
| | KA | TA | CA | KA | TA | CA | KF |
| Norm only | 80.25 | 18.65 | 46.00 | 76.96 | 17.59 | 27.64 | 7.28 |
| Direction only | 77.64 | 27.91 | 50.55 | 75.58 | 29.93 | 31.32 | 8.66 |
| PCA | 78.89 | 23.18 | 47.83 | 76.26 | 29.16 | 27.91 | 7.98 |
| Random | 80.14 | 25.51 | 45.30 | 76.66 | 27.93 | 29.59 | 7.58 |
| Ours | 79.20 | 21.11 | 56.87 | 80.77 | 31.03 | 34.81 | 3.47 |

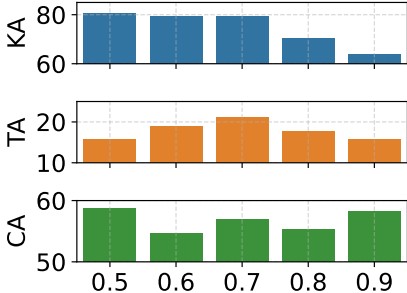

Figure 11: Boundary between prototype and classifiers

### E.3 DIFFERENT THRESHOLDS FOR THRESHOLDING METHODS

In our main text, we compare several methods that utilize threshold-based approaches, for which we provide an optimized threshold selection. In Figs. 12, 13, 14 and 15, we present a comparison of threshold selection across different methods alongside our results. The findings indicate that competing methods are highly sensitive to threshold choices, making it difficult to achieve both high TA and high CA simultaneously. In contrast, our method maintains a better balance between TA and CA, demonstrating greater stability and robustness across different threshold settings.

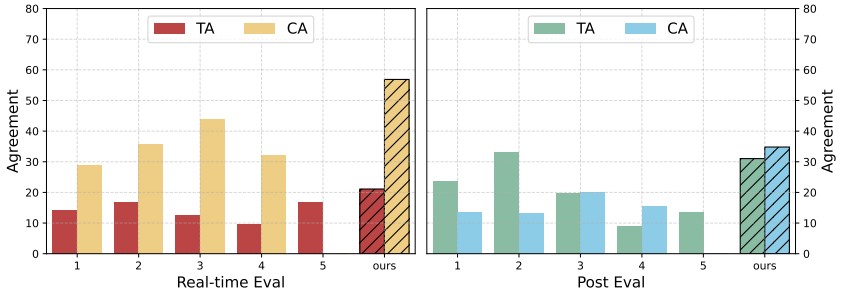

Figure 12: Euclidean threshold

### E.4 RUNNING TIME

We evaluate the running time of the compared methods. The results are shown in Fig. 16, which show that training-required methods require significantly more time, while training-free approaches are generally faster. Our method takes longer than threshold-based methods but remains faster than training-based approaches. Additionally, incorporating the SC strategy further increases running time.

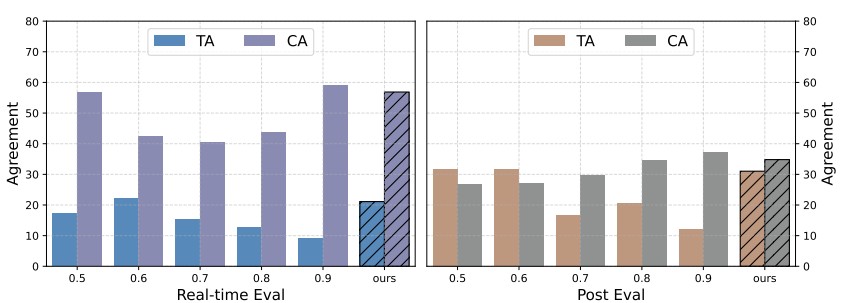

Figure 13: Cosine threshold

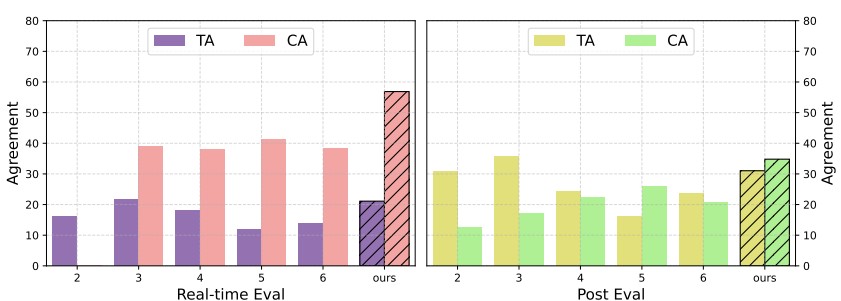

Figure 14: Magnitute threshold

### E.5 COMPARISONS WITH EXISTING METHODS UNDER LONG-TAILED TTD SETTING

In Table 5, we provide the results of our method under long-tailed TTD task. In Table 15, we further investigate the comparisons with existing methods under the same long-tailed TTD setting. Specifically, to conduct the experiment, the 30 novel classes of CIFAR-100D at test time are distributed with significant imbalance. Specifically, some novel classes are under-represented, while others have many more samples. This simulates real-world data drift where new concepts may appear unequally in the stream. The remaining 70 known classes follow the standard uniform distribution. All test samples (known + novel) are randomly shuffled and sequentially fed to the model during evaluation, ensuring that the online nature of TTD is preserved. Compared to the balanced distribution, we observe that our performance competitive under long-tailed distributions, which validates our memory-based discovery method's robustness. These results suggest that our approach remains effective even when novel class number is imbalanced.

Table 15: Comparisons with existing methods under Long-tailed TTD setting.

|  | **Real-time Eval** | | | **Post Eval** | | | |
|---|---|---|---|---|---|---|---|
|  | KA | TA | CA | KA | TA | CA | KF |
| Unknown classes long-tail(GMP) | 75.26 | 8.90 | 39.38 | 70.16 | 12.31 | 20.52 | 15.23 |
| Unknown classes long-tail(PHE) | 72.37 | 16.24 | 36.42 | 69.24 | 14.63 | 24.77 | 3.78 |
| Unknown classes long-tail(Ours) | 78.41 | 24.69 | 54.13 | 78.69 | 27.51 | 34.37 | 5.55 |
| Randomly (Ours) | 79.20 | 21.11 | 56.87 | 80.77 | 31.03 | 34.81 | 3.47 |

### E.6 RUNTIME AND MEMORY BEHAVIOR UNDER STREAMING

We report the runtime and memory consumption under different hyperparameter settings in Table 16. The results show that latency grows smoothly with memory size $K$, bucket neighbors, and top-$k$ comparisons, while memory usage remains effectively constant across all settings. No configuration introduces abrupt runtime growth, confirming that HM imposes mild, controllable overhead throughout streaming TTD.

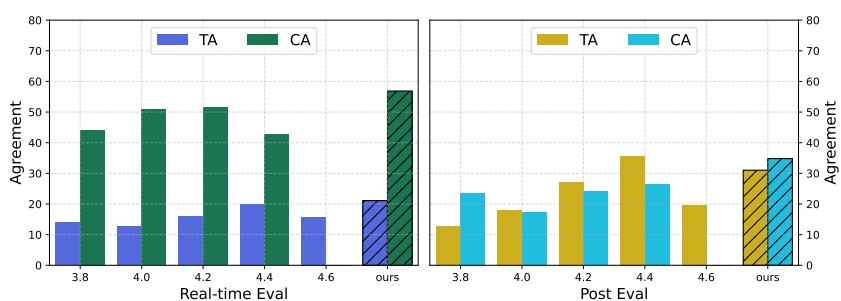

Figure 15: Entropy threshold

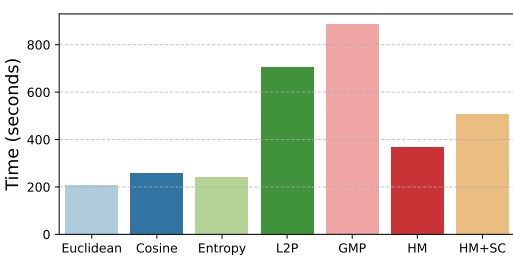

Figure 16: Comparisons of running time.

## F RESULTS ON MORE DATASETS

### F.1 ADDITIONAL RESULTS ON STANFORD CARS

To further examine scalability under fine-grained categories, we additionally evaluate our method on the Stanford Cars dataset (Krause et al., 2013). This dataset contains 196 classes with subtle inter-class differences, which makes test-time discovery more challenging than on coarse-grained benchmarks. We construct a dataset split with 140 known and 56 unknown classes. The results are presented in Table 17. We observe that HM maintains consistent advantages over GMP and PHE in both real-time and post-hoc evaluations, indicating that the proposed memory-based discovery mechanism remains effective even on highly fine-grained streams.

### F.2 LARGE-SCALE OPEN-WORLD EVALUATION

To evaluate the scalability of HM in substantially larger open-world environments, we construct an ImageNet-derived test-time stream that contains more than 1,000 object categories. Specifically, we use a pretrained model from Tiny-ImageNet (200 classes) and evaluate it on the full ImageNet-resized test set. The number of unseen categories is varied from 200 to 800 to simulate progressively more challenging open-world conditions. Importantly, the model is not retrained or adapted, and only TTD is performed, fully aligned with the TTD setting. Table 18 reports results for GMP, PHE, and HM under both real-time and post-hoc evaluation protocols. Across all scales, HM consistently achieves higher CA and TA than existing methods, indicating stronger novel-class discovery performance. The improvements remain stable even when the number of novel categories increases fourfold (from 200 to 800), demonstrating that HM maintains robust discovery behavior under large-scale open-world streams. These large-scale experiments confirm that (1) HM scales effectively to long streams containing more than 1,000 categories, (2) performance degradation remains mild even as the number of unseen classes increases dramatically, and (3) the proposed memory-based discovery mechanism preserves its advantages over both GMP and PHE across all difficulty levels. This demonstrates that HM is well-suited for realistic open-world scenarios that require high-capacity novel-class discovery during test time.

Table 16: Streaming-time latency and memory usage under different hyperparameter settings.

| Stored samples / class | | | Norm discretization $\kappa$ | | | # Buckets | | | Top-$k$ compared | | |
|---|---|---|---|---|---|---|---|---|---|---|---|
| Value | Time(s) | MB | Value | Time(s) | MB | Value | Time(s) | MB | Value | Time(s) | MB |
| 0 | 484.24 | 8799 | 1 | 552.14 | 8865 | 0 | 362.38 | 8863 | 0 | 452.60 | 8863 |
| 10 | 495.51 | 8831 | 2 | 505.55 | 8863 | 2 | 489.73 | 8863 | 5 | 483.45 | 8863 |
| 20 | 505.55 | 8863 | 4 | 481.17 | 8863 | 4 | 536.21 | 8863 | 10 | 505.55 | 8863 |
| 30 | 521.59 | 8895 | 10 | 486.84 | 8862 | 6 | 566.45 | 8863 | 15 | 520.11 | 8863 |

Table 17: Results on Stanford Cars.

| Method | Real-Time Eval | | | Post-Hoc Eval | | | |
|---|---|---|---|---|---|---|---|
| | KA | TA | CA | KA | TA | CA | KF |
| GMP | 25.05 | 15.45 | 21.98 | 20.45 | 12.68 | 23.36 | 7.99 |
| PHE | 28.44 | 16.82 | 25.22 | 23.85 | 14.69 | 24.25 | 5.43 |
| Ours | **30.24** | **18.16** | **34.72** | **26.24** | **15.98** | **28.38** | **3.77** |

Table 18: Performance on the ImageNet-derived open-world streams (200 to 800 unseen classes). The model is trained on 200 classes from Tiny-ImageNet and evaluated on ImageNet-resized with varying numbers of unseen classes.

| #Unknown | Method | Real-Time Eval | | | Post-Hoc Eval | | | KF |
|---|---|---|---|---|---|---|---|---|
| | | KA | TA | CA | KA | TA | CA | ($\downarrow$) |
| 200 | GMP | 65.67 | 16.14 | 46.57 | 64.80 | 10.88 | 27.10 | 10.50 |
| | PHE | 67.75 | 16.29 | 50.94 | 66.27 | 13.24 | 28.61 | 7.59 |
| | Ours | **70.93** | **18.47** | **55.78** | **71.28** | **16.07** | **33.74** | **3.62** |
| 400 | GMP | 63.58 | 13.41 | 45.76 | 61.15 | 7.81 | 20.54 | 11.36 |
| | PHE | 65.11 | 14.19 | 47.32 | 64.53 | 9.63 | 24.23 | 8.26 |
| | Ours | **68.09** | **16.77** | **51.85** | **67.50** | **14.31** | **30.16** | **5.86** |
| 600 | GMP | 58.49 | 10.29 | 37.69 | 57.98 | 6.03 | 18.15 | 14.52 |
| | PHE | 60.33 | 11.80 | 40.26 | 59.85 | 9.04 | 24.61 | 10.15 |
| | Ours | **64.42** | **14.51** | **46.56** | **63.79** | **11.44** | **27.60** | **8.32** |
| 800 | GMP | 55.56 | 8.37 | 30.97 | 53.50 | 5.34 | 14.76 | 18.75 |
| | PHE | 58.35 | 9.18 | 36.20 | 55.03 | 7.31 | 18.12 | 16.00 |
| | Ours | **62.92** | **12.54** | **43.25** | **60.39** | **10.74** | **25.00** | **10.69** |

## G  CLARIFICATION ON REAL-TIME EVALUATION AND COMPUTATIONAL COMPLEXITY

**Real-time evaluation.** In this work, the term *real-time evaluation* refers to the evaluation protocol used in Test-Time Discovery (TTD), in which each incoming sample must be classified immediately without revisiting previously seen data. This contrasts with the post-hoc evaluation paradigm commonly adopted in NCD, GCD, and OCD. The term therefore characterizes the evaluation setup rather than implying strict system-level latency guarantees.

**Computational complexity.**  For each test input, the overall computational cost of HM is $O\left(d(R + S)\right)$, where $d$ denotes the feature dimensionality, $R$ is the hash code length, and $S$ is the bounded number of stored samples retrieved from the assigned bucket and its neighboring buckets. Hashing contributes $O(dR)$, bucket-level comparisons contribute $O(dS)$, and the amortized cost of self-correction remains constant. Since both $R$ and $S$ are small fixed constants, the per-sample inference time does not increase as the stream grows or as more classes are discovered, thus supporting real-time evaluation under the TTD protocol.

**Memory complexity.** The memory footprint of HM is $O(CKd)$, where $C$ is the total number of known and discovered classes, $K$ is the capped number of stored samples per class, and $d$ is the feature dimension. Importantly, retrieval cost is independent of the total memory size, as each query

only touches one bucket and a small number of neighboring buckets. This property allows HM to scale effectively to long streaming scenarios while maintaining efficient discovery performance.

## H  LLM USAGE DISCLOSURE

We used LLMs solely to correct grammatical errors in the writing. The model was not involved in research design, data analysis, or result generation.

