# OpenReview forum: "Real-Time Evaluation for Novel Class Discovery at Test Time"
_ICLR.cc/2026/Conference — Submitted to ICLR 2026_

### Official Review · Reviewer_ucGd · 2025-10-24

**Soundness:** 2
**Presentation:** 2
**Contribution:** 2
**Rating:** 2
**Confidence:** 3

**Summary:**

This paper proposes the task of test-time discovery, i.e. generalized category discovery (classifying known categories and identify new ones) where at test time data is processed online, making clustering more difficult. To tackle this task the paper proposes a hash memory method that uses locality-sensitive hashing and a global-to-local strategy for efficient retrieval and novel class formation, with a lightweight self-correction mechanism.

**Strengths:**

- The focus on a online testing setting for novel class discovery is interesting and makes the problem more suitable for different scenarios
***

- The experiments exploring the effect of the number of unknown classes and different test-time distributions are appreciated. this analysis adds some empirical depth beyond a single evaluation setup.

**Weaknesses:**

- relation to prior work and conceptual clarity
- The paper does not sufficiently situate its contribution within prior work. It is unclear whether similar approaches have already been explored in the GCD or related literature. The related work sections and problem definition settings do attempt to do this but it remains unclear
-  For instance, the intro states "In TTD, the model must not only classify known categories but also decide in real time whether a sample belongs to a previously discovered class or should initiate a new one.". This is exactly what the problem of generalized category discovery addresses. This is also not mentioned when GCD is described in the related work. It is made clearer in the comparison in Table 1 but should be clear from the start.
- The relation to PHE (Zheng et al., 2024) is unclear. That work also claims an online or real-time discovery setting, in contrast to how it is represented in Table 1. It is also missing from the related work despite clear conceptual overlap.
- PHE is compared to in the experiments, however the conceptual difference should be made clear
***

- comparisons to state-of-the-art
- related to the above point since the main difference between GCD and the proposed task is the online testing it seems that GCD methods should be compared to, at least in the post evaluation setting
- it would also strengthen the papers claims to compare to GCD methods in the online setting by iteratively testing as the test set expands to demonstrate that GCD methods are either much worse in this setting or too computationally expensive.
***

- experimental design and fairness
- The choice of datasets and the 7:3 known–unknown class ratio is not well justified. Given the similarity to GCD or on the fly category discovery (OCD), it would be more convincing to adopt standard datasets and splits used in priors works. Some of the datasets are the same but it isn't clear why tiny-imagenet is chosen over imagenet-100 and why Stanford cars isn't used. Prior works also tend to use a 1:1 ratio of known to unknown classes.
- The new evaluation metrics introduced in the paper seem to obscure rather than clarify the relationship between GCD/OCD and the proposed setting. While some new evaluation is reasonable, several metrics appear to be renamed or slightly altered versions of existing GCD/OCD metrics, making interpretation difficult.
***

- ablation study
- The ablation study is limited, evaluating only the presence of the hash memory and self-correction components. Key elements such as the graph-augmented retrieval or the global-local hybrid strategy are not analyzed.
- The reported ablation results are unconvincing. While cluster agreement improves, both true-label agreement and known-class accuracy (KA) decline. With the issue mentioned above, its hard to interpret these metrics and weight which is more important and evaluate whether the proposed components are actually benefitting the model.
***

- presentation
- The paper relies heavily on acronyms, which makes the results and reasoning difficult to follow in places.

**Questions:**

- How does your approach relate to PHE (Zheng et al., 2024), which also performs online or real-time discovery? The distinction is unclear from both the related work and Table 1.
***

- How does the proposed test-time discovery (TTD) setting fundamentally differ from Generalized Category Discovery (GCD) beyond the sequential or streaming data assumption? Why are GCD baselines not included in the post evaluation comparisons?
***

- Why were the chosen datasets and the 7:3 known–unknown ratio used? Would the method still perform well on standard GCD datasets and splits?
***

- Several of the proposed evaluation metrics seem to be renamed or modified versions of standard ones. Can you clarify the link between existing metrics and new metrics and justify why new terminology was introduced instead of extending existing metrics?
***

- The ablation omits analysis of several important components (e.g., graph-augmented retrieval, global-local hybrid strategy). Could you provide further justification or additional experiments?
***

- In Table 7, cluster agreement improves but true-label agreement and KA decrease. What does this trade-off imply about the quality of the discovered clusters?

---

> ### Author Response · Authors · 2025-11-23
>
> ### W1&(Q1,Q2) Relation to prior work and conceptual clarity
>
> We thank the reviewer for the helpful comments. We respond to each point below.
>
> (1) **Insufficient positioning within prior GCD/OCD literature**： TTD builds on discovery tasks such as NCD, GCD, and OCD but differs in one key aspect: prior works separate discovery from testing (post-hoc clustering or full test-set assessment), whereas TTD requires discovery and predict to occur simultaneously on a streaming test-time sequence (sample-level or batch-level). This constraint fundamentally changes how a model must operate, as it cannot rely on access to future samples or post-hoc correction. We will clarify this in the revision.
>
> (2) **Clarify the distinction between TTD and GCD from the start**: We agree that the current phrasing may appear similar to GCD if the evaluation protocol is not stated upfront. The key distinction is the timing of decisions. In GCD, all test samples are available simultaneously and labels are assigned after performing global clustering on the full set. There is no notion of “earlier” or “later” predictions, and no risk of a cluster initially treated as novel later being absorbed into a known class, because the full data distribution is visible at once. In TTD, however, each test sample must be classified immediately, without access to future samples and without any opportunity for global reassignment afterward (such as in auto drivings). This makes temporal consistency and irreversible decisions central to the problem, and creates challenges that do not exist in GCD. Table 1 highlights this distinction, and we will revise the introduction and related work to state this difference explicitly from the beginning.
>
> (3) **Clarify conceptual relation between TTD and PHE**: We appreciate this comment and will emphasize the conceptual distinction. PHE performs on-the-fly category discovery (OCD) in its training/processing stage, but its **test and evaluation is still post-hoc**: it assumes access to the full test set for clustering or label assignment after processing ends. Under true test-time streaming, where labels must be assigned on arrival and cannot be revised, PHE is not directly applicable. Our experiments include PHE adapted to this stricter setting, and its drop in performance reflects precisely this mismatch.
>
> (4) **Explicitly differentiate HM from PHE beyond experimental comparison**: We agree, and we will explicitly state that the comparison is meant to show that methods designed for post-hoc evaluation degrade when forced into a true test-time scenario. Our framework is specifically designed for this setting: it enables simultaneous known-class prediction and novel-class discovery during testing without access to all past and future samples. This capability is not supported by existing NCD/GCD/OCD methods by design.

---

> ### Author Response · Authors · 2025-11-23
>
> ### W2&Q2 Comparisons to GCD methods
>
> We thank the reviewer for the valuable suggestions. We address both subpoints below.
>
> (1) **GCD methods should be compared**: GCD's evaluation protocol fundamentally differs from TTD. GCD assumes two disjoint datasets, where one for discovery and one for evaluation, and requires access to the entire test set for post-hoc clustering. TTD instead performs discovery and evaluation simultaneously on the same streaming test set, assigning each sample immediately without seeing future data. Because GCD relies on full-dataset global clustering, it cannot operate under this streaming constraint. Running GCD in its native post-eval setting would not reflect TTD’s real-time prediction requirements.
>
> (2) **Compare GCD methods in an online setting as the test set expands**: We appreciate the suggestion. However, adapting GCD to an online-TTD regime is not feasible without violating its core assumptions. GCD critically depends on global clustering: restricting clustering to each arriving batch destroys the global structure required to maintain class identity over time. With only batch-level clusters, a model cannot determine whether a cluster in batch $t$ corresponds to a previously discovered class or represents a new one, and the ambiguity becomes severe as batch sizes shrink. In the extreme case of batch size 1, GCD methods degenerate entirely because clustering is not possible.
>
> To examine this point empirically, we followed the reviewer’s suggestion and built a GCD baseline by replacing global clustering stage with local batch-wise clustering applied incrementally on the streaming test data.
> The results are shown below.
> As predicted by the online literature, the adapted variant collapses immediately: the discovered categories become unstable, rapidly drift, and fail to maintain consistent identities across batches. This confirms that the poor performance is not due to implementation details but reflects a fundamental incompatibility between GCD’s global clustering requirement and the constraints of test-time streaming.
>
> |      | REAL  |       |       | POST  |       |       |       |
> | ---- | ----- | ----- | ----- | ----- | ----- | ----- | ----- |
> |      | KA    | TA    | CA    | KA    | TA    | CA    | KF    |
> | GCD  | 70.94 | 8.70  | 10.95 | 71.85 | 9.51  | 11.45 | 12.30 |
> | Ours | 79.20 | 21.11 | 56.87 | 80.77 | 31.03 | 34.81 | 3.47  |
>
> For these reasons, OCD-style methods such as PHE, designed to operate online, serve as the appropriate baselines. Our experiments already include PHE in the strict test-time streaming setting, where its performance drops substantially, illustrating that post-hoc discovery approaches are not robust under TTD constraints. The post-hoc results in our paper are included only to demonstrate compatibility. The primary focus of TTD is real-time prediction and evaluation rather than offline assessment.
>
> ### W3&Q3 Experimental design and fairness
>
> #### W3.1 Experimental design
>
> We thank the reviewer for the insightful comments. We address the dataset and ratio concerns as follows.
>
> (1) **The choice of datasets and the 7:3 ratio is not well justified**:
> The TTD setting differs from NCD/GCD/OCD in a critical way: discovery and test occur on the *same streaming test set*, rather than using two separate sets as in prior works. This means that standard dataset splits from GCD or OCD cannot be directly reused because their discovery-test separation does not apply under TTD. In TTD, unknown classes appear *within* the test stream, making the problem substantially more challenging than NCD/GCD/OCD. For this reason, we adopt a 7:3 ratio (known classes are more than unknowns) as the primary setting to ensure more known knowledge can be learned before test.

---

> ### Author Response · Authors · 2025-11-23
>
> (2) **Why not ImageNet-100 or Stanford Cars?**:
> Our selection covers four widely used benchmarks (CIFAR100, CUB, Aircraft, Tiny-ImageNet), spanning coarse-grained, fine-grained, and medium-scale datasets. Following the reviewer’s suggestion, we additionally evaluated models trained on Tiny-ImageNet (200 classes) and tested on ImageNet-resized (200+800 classes), as well as on Stanford Cars. These results yield the same conclusions and will be included in the revision.
>
> First, to evaluate HM under larger-scale open-world conditions, we conducted additional experiments using an ImageNet-derived stream with more than 1,000 categories. To avoid retraining large models and focus specifically on test-time scalability, we used a model pretrained on Tiny-ImageNet (200 classes) and evaluated it on the full ImageNet-resized test set (200 + 800 unknown), varying the number of unknown categories from 200 to 800.
> The results are shown below:
>
> | #Unknown | REAL-TIME |       |       | POST-HOC |       |       |       |
> | -------- | --------- | ----- | ----- | -------- | ----- | ----- | ----- |
> | 200      | KA        | TA    | CA    | KA       | TA    | CA    | KF    |
> | GMP      | 65.67     | 16.14 | 46.57 | 64.80    | 10.88 | 27.10 | 10.50 |
> | PHE      | 67.75     | 16.29 | 50.94 | 66.27    | 13.24 | 28.61 | 7.59  |
> | Ours     | 70.93     | 18.47 | 55.78 | 71.28    | 16.07 | 33.74 | 3.62  |
> | 400      |           |       |       |          |       |       |       |
> | GMP      | 63.58     | 13.41 | 45.76 | 61.15    | 7.81  | 20.54 | 11.36 |
> | PHE      | 65.11     | 14.19 | 47.32 | 64.53    | 9.63  | 24.23 | 8.26  |
> | Ours     | 68.09     | 16.77 | 51.85 | 67.50    | 14.31 | 30.16 | 5.86  |
> | 600      |           |       |       |          |       |       |       |
> | GMP      | 58.49     | 10.29 | 37.69 | 57.98    | 6.03  | 18.15 | 14.52 |
> | PHE      | 60.33     | 11.80 | 40.26 | 59.85    | 9.04  | 24.61 | 10.15 |
> | Ours     | 64.42     | 14.51 | 46.56 | 63.79    | 11.44 | 27.60 | 8.32  |
> | 800      |           |       |       |          |       |       |       |
> | GMP      | 55.56     | 8.37  | 30.97 | 53.50    | 5.34  | 14.76 | 18.75 |
> | PHE      | 58.35     | 9.18  | 36.20 | 55.03    | 7.31  | 18.12 | 16.00 |
> | Ours     | 62.92     | 12.54 | 43.25 | 60.39    | 10.74 | 25.00 | 10.69 |
>
> Across all scales (200 to 800 unknown classes), our method outperforms GMP and PHE, indicating that the proposed memory-based discovery mechanism remains stable even when the number of novel categories grows fourfold.
> These results suggest that HM generalizes well to large-scale streams and that its memory-based discovery framework retains robustness even under significantly expanded open-world settings. We will include these results in our paper.
>
> Second, we evaluate on Standford Cars dataset. The results are shown below. We find that our method can also achieve better performance than GMP and PHE.
>
>
> | 140+56 | REAL-TIME |       |       | POST-HOC |       |       |      |
> | ------ | --------- | ----- | ----- | -------- | ----- | ----- | ---- |
> |        | KA        | TA    | CA    | KA       | TA    | CA    | KF   |
> | GMP    | 25.05     | 15.45 | 21.98 | 20.45    | 12.68 | 23.36 | 7.99 |
> | PHE    | 28.44     | 16.82 | 25.22 | 23.85    | 14.69 | 24.25 | 5.43 |
> | Ours   | 30.24     | 18.16 | 34.72 | 26.24    | 15.98 | 28.38 | 3.77 |
>
> (3) **Prior works use a 1:1 known–unknown ratio**:
> We thank the reviewer for noting this. Our paper already includes additional ratios (such as 8:2 and 9:1 in Table 3 (updated 4)). In response to the reviewer’s request, we further provide experiments with the 1:1 split (50 known + 50 unknown). As shown below, our method consistently improves TA/CA across all methods.
>
> | Known+Unknown | REAL-TIME |       |       | POST-HOC |       |       |       |
> | ------------- | --------- | ----- | ----- | -------- | ----- | ----- | ----- |
> | 50+50         | KA        | TA    | CA    | KA       | TA    | CA    | KF    |
> | GMP           | 70.95     | 21.95 | 30.07 | 68.45    | 15.58 | 28.49 | 11.55 |
> | PHE           | 72.86     | 25.04 | 33.68 | 70.10    | 22.16 | 31.68 | 7.81  |
> | Ours          | 75.62     | 33.70 | 40.21 | 76.24    | 28.26 | 38.00 | 4.62  |
>
> **About the fairness: All methods in our experiments are evaluated under identical conditions to ensure fairness.**

---

> ### Author Response · Authors · 2025-11-23
>
> #### W3.2&Q4 Metrics
>
> We thank the reviewer for the helpful comment. The intent of our evaluation design is not to rename or slightly modify existing NCD/GCD/OCD metrics, but to address a fundamental incompatibility between those metrics and the TTD setting. GCD and OCD both rely on post-hoc evaluation: cluster purity or clustering accuracy is computed only after the *entire* test set is available. Such metrics cannot be computed in TTD because the model must assign labels immediately for each incoming sample and future samples are not observable. Therefore, classical clustering-based metrics are not definable in the test-time streaming regime.
>
> Our agreement-based metrics are the natural adaptation of these post-hoc measures to the TTD constraint. Instead of requiring access to all data at once, they accumulate the same notion of correctness over time, enabling real-time assessment. The design follows directly from the need to evaluate discovery and evaluation *as they occur*, without violating the streaming protocol. Detailed definitions and their connections to standard metrics are provided in Appendix C.
>
> ### W4 Ablation study
>
> #### W4.1&Q5 Ablation study of graph-augmented retrieval and global-local hybrid strategy
>
> We would like to clarify that the components mentioned by the reviewer have already been analyzed in Appendix.
>
> (1) **Graph-augmented retrieval (Appendix D.1 & D.2).**:
> We provide detailed studies on (a) the number of neighbor buckets in the dynamic graph (Fig. 9) and (b) the number of compared samples $k$ inside each bucket (Table 9 (updated 10)). Both experiments show clear and interpretable effects: too few neighbors underutilize structure while too many introduce noise, revealing the necessity of graph smoothing for stable discovery. These analyses directly quantify the role of the graph in TTD performance.
>
> (2) **Global–local hybrid strategy (Appendix E.2).**:
> We also analyze the boundary that switches between prototype-based classification (global) and LSH-based retrieval (local), as shown in Fig. 11. Increasing the threshold favors local retrieval and sharply degrades KA, while decreasing it destabilizes discovery. This demonstrates that the hybrid mechanism is not heuristic but critical to balancing known-class stability and novel-class discovery in TTD.
>
> #### W4.2&Q6 Why CA improves while TA and KA decline?
>
> In the TTD setting, evaluation naturally decomposes into two parts: known class recognition and novel class discovery. For novel classes, our goal is not only to align clusters with ground truth labels but also to maintain stable cluster identities over time in a streaming scenario.
>
> The three metrics capture complementary aspects of this alignment. CA measures how pure each discovered cluster is with respect to the true labels, while TA measures how concentrated each true class is within the discovered clusters. An increase in CA together with a mild decrease in TA therefore indicates that samples assigned to the same discovered category become more label consistent, at the cost of slightly splitting some true classes across multiple discovered categories. Under online TTD constraints, this trade off is often preferable, since over merging different classes into a single cluster is much more harmful for long term discovery than conservatively splitting a class into a few stable clusters.
>
> KA focuses on known class classification and can be temporarily affected by the progress of discovery and the presence of borderline samples, which explains its slight drop in some ablations.
>
> However, across the full model (not extreme ablations), the components improve the joint CA–TA behavior that determines discovery reliability throughout the stream. We will clarify in the revision that the goal of these ablations is to reveal the inherent trade-offs in TTD rather than to enforce simultaneous increases across all metrics.
>
> ### W5 Acronyms suggestion
>
> We thank the reviewer for this helpful observation. In the revised manuscript, we will reduce unnecessary abbreviations, expand all key terms upon first appearance, and include a small acronym table for quick reference. We believe these changes will significantly improve clarity and presentation.

---

> > ### Comment · Reviewer_ucGd · 2025-11-26
> >
> > Thank you for the response. I particularly appreciate the additional experiments and feel my concerns about the relation to prior work, comparison to GCD method's, known:unknown ratio and dataset choice have been satisfied. Regarding the ablations, I appreciate the pointers to Appendix D1, D2 and E2. These satisfy my concern around the missing ablations.
> >
> > I am however still concerned about the unconvincing ablation results in Table 7. I do appreciate the explanation of the metric trade-offs given in the rebuttal, however a model that improves all metrics would naturally be more convincing than the proposed model which has a positive effect on one (CA) metric and a negative effect on two (TA, KF). Maybe there is some misunderstanding here as the same experiment is listed with positive KF values in some Tables (e.g. 5) and negative in others (e.g. 7) so its hard to tell if lower is better for this metric in Table 7 as indicated by the arrows in Table 1. Having a trade-off in metrics is fine, my main concern is that the baseline model without the proposed components HM and SC would already be better than the prior works compared to (i.e. comparing row 1 of Table 7 to the results in Table 1). I'd appreciate it if the authors could clarify what exactly row 1 of Table 7 is and why its better than the prior works on most metrics.
> >
> > My only other remaining concern is the paper presentation. While the authors have promised to improve this they didn't take the opportunity to submit a revised paper to demonstrate this and the rebuttal contained the same heavy overuse of acronyms as the paper making it quite hard to understand. This isn't a reason to reject the paper but it would certainly help future readers.
> >
> > I currently intend to raise my rating by one step as most my concerns have been addressed, however I still have concerns about the ablation in Table 7 so cannot confidently recommend accept.

---

> ### Author Response · Authors · 2025-11-28
>
> Thank you for the follow-up. We appreciate that most concerns have been resolved. Below, we address the two remaining issues and confirm that all promised revisions have been incorporated into the updated manuscript.
>
> **1.Clarification on Table 7 (updated 8) Ablation**
>
> **(a) KF sign issue**: The positive/negative KF inconsistency was a notation error. We have unified the definition to
> $\text{KF} = \text{initial accuracy} - \text{final accuracy}$, so all KF values are now non-negative and consistent across all tables. The corrected values appear in Tables 6 and 7 (updated 7 and 8) of the revised manuscript.
>
> **(b) Why the baseline in Table 7 (updated 8) appears stronger than prior methods**: Row 1 of Table 7 is not a prior method. It is our internal ablation backbone containing the same ViT backbone plus exponential moving average (EMA) refinement but without the two components HM or SC. Prior works in Table 1 use their own pipelines and do not include EMA. This difference is now explicitly stated in Sec. 5.4. The results indicate that EMA, although simple, provides a meaningful benefit under TTD.
>
> **2. Presentation Improvements**: We have reduced unnecessary acronyms, expanded key terms at first appearance, and simplified wording in the paper. These changes have been integrated into the revised manuscript to ensure improved clarity for future readers. For example, we only keep the acronyms TTD, NCD, and HM in the Introduction section.
>
> **3. Revised Manuscript Submitted**: The textual changes, corrected tables, and additional experiments have been integrated into the submitted revised manuscript.
>
> We hope these clarifications fully resolve the remaining concerns.

---

### Official Review · Reviewer_vbHh · 2025-10-31

**Soundness:** 3
**Presentation:** 3
**Contribution:** 3
**Rating:** 6
**Confidence:** 4

**Summary:**

This paper introduces Test-Time Discovery (TTD), a new protocol for real-time novel class discovery that requires models to simultaneously classify known categories and identify emerging ones under sequential test-time constraints. The authors propose a training-free Hash Memory (HM) framework that combines semantic-aware hashing of feature norms and directions, a cooperative inference strategy integrating global prototypes and memory-based reasoning, and a self-correction mechanism to refine mislabeled samples. Experimental results across multiple benchmarks show that HM achieves more accurate and stable real-time discovery than prior NCD and TTT methods while maintaining strong performance on known classes.

**Strengths:**

- First of all, the paper introduces Test-Time Discovery (TTD), a sequential real-time evaluation protocol that bridges the gap in existing NCD approaches by jointly measuring classification and discovery.
- The proposed Hash Memory combines semantic-aware hashing, graph-augmented retrieval, and self-correction into a simple yet effective training-free design for real-time novel class discovery. Especially, a hybrid strategy integrates a global prototype classifier for confident known samples with an LSH-based local voting mechanism for ambiguous or novel ones, ensuring stability on known classes while enabling flexible discovery.
- Comprehensive experiments analyze real-time vs. post-hoc metrics (KA, KF, TA, CA), parameter sensitivity, and robustness under varied memory sizes, sample orders, and data distributions, consistently outperforming NCD and TTT baselines across benchmarks.

**Weaknesses:**

- A first concern is an inherent ambiguity in the term real-time evaluation. It is unclear whether it refers to quantitative latency and memory overheads for hashing, neighbor retrieval, and SC updates.
- The system relies on multiple heuristic mechanisms, making it difficult to clearly attribute performance gains to individual modules and to isolate their true effectiveness within the overall framework.
- The provided results are limited to relatively short sequential test streams with a small number of discovered classes, leaving it unclear how the method would perform under long-term discovery or class reoccurrence which are central challenges in realistic continual open-world scenarios.

**Questions:**

- How does it estimate or calibrate prediction confidence to prevent early pseudo-label errors from propagating?
- What is the computational and memory complexity of the hash-based graph retrieval and self-correction steps as the number of discovered classes grows, and can the method remain real-time in longer streams?

---

> ### Author Response · Authors · 2025-11-23
>
> ### W1&Q2 An inherent ambiguity in the term real-time evaluation
>
> We appreciate the reviewer’s concern. In our paper, “real-time evaluation” refers to the evaluation protocol for TTD, where each test sample must be classified immediately without revisiting past data, in contrast to classic NCD/GCD/OCD settings that rely on post-hoc discovery and evaluation. It is a property of the evaluation setup rather than a claim about strict system-level latency benchmarks.
>
> We appreciate the reviewer’s question regarding the scalability of the hash-based retrieval and self-correction components.
> The overall computational complexity of HM for each test sample is $O(d(R+S))$, where $d$ is the feature dimensionality, $R$ is the size of the hash code, and $S$ is the bounded number of stored samples compared during retrieval (from the assigned bucket and its graph neighbors). Hashing contributes $O(dR)$ and neighbor comparisons contribute $O(dS)$, while the amortized cost of self-correction remains constant. Because $R$ and $S$ are small fixed constants, the per-sample computation does not increase as more classes are discovered or as the stream becomes longer, allowing the method to remain real-time.
>
> The memory complexity is $O(CKd)$, where $C$ is the number of knwon and discovered classes, $K$ is the maximum number of stored samples per class, and $d$ is the sample size. Memory grows linearly with the number of classes, but retrieval cost is independent of total memory size, since each query touches only one bucket and a few neighboring buckets. This ensures that HM scales gracefully to long streams while maintaining efficient discovery behavior.
>
> ### W2 Difficult to clearly attribute performance gains to individual modules
>
> We thank the reviewer for the insightful comment. We would like to clarify that HM is not a collection of independent heuristic mechanisms, but a unified framework built around a hash-based memory designed specifically for the requirements of TTD. The core methodology consists of three tightly coupled components: (1) constructing a memory that organizes features into stable buckets, (2) performing prediction and discovery through efficient local retrieval within this memory, and (3) applying lightweight self-correction to mitigate the impact of early discovery errors. These elements are not optional heuristics but are driven directly by TTD’s demands for online, non-revisitable decisions without model updates.
>
> Importantly, the framework enables a model trained solely on known classes to perform simultaneous known-class prediction and novel-class discovery during test time—capabilities that existing NCD, GCD, and OCD settings do not support. This task-driven formulation, together with the targeted ablations reported in the paper, demonstrates that the performance gains arise from the structured design of the memory-based TTD pipeline rather than from ad-hoc heuristic combinations. We will make this motivation clearer in the revision.

---

> ### Author Response · Authors · 2025-11-23
>
> ### W3 Long-term open-world streams
>
> We thank the reviewer for raising this important question about scalability.
> To evaluate HM under larger-scale open-world conditions, we conducted additional experiments using an ImageNet-derived stream with more than 1,000 categories. To avoid retraining large models and focus specifically on test-time scalability, we used a model pretrained on Tiny-ImageNet (200 classes) and evaluated it on the full ImageNet-resized test set (200 + 800), varying the number of unknown categories from 200 to 800.
> The results are shown below:
>
> | #Unknown | REAL-TIME |       |       | POST-HOC |       |       |       |
> | -------- | --------- | ----- | ----- | -------- | ----- | ----- | ----- |
> | 200      | KA        | TA    | CA    | KA       | TA    | CA    | KF    |
> | GMP      | 65.67     | 16.14 | 46.57 | 64.80    | 10.88 | 27.10 | 10.50 |
> | PHE      | 67.75     | 16.29 | 50.94 | 66.27    | 13.24 | 28.61 | 7.59  |
> | Ours     | 70.93     | 18.47 | 55.78 | 71.28    | 16.07 | 33.74 | 3.62  |
> | 400      |           |       |       |          |       |       |       |
> | GMP      | 63.58     | 13.41 | 45.76 | 61.15    | 7.81  | 20.54 | 11.36 |
> | PHE      | 65.11     | 14.19 | 47.32 | 64.53    | 9.63  | 24.23 | 8.26  |
> | Ours     | 68.09     | 16.77 | 51.85 | 67.50    | 14.31 | 30.16 | 5.86  |
> | 600      |           |       |       |          |       |       |       |
> | GMP      | 58.49     | 10.29 | 37.69 | 57.98    | 6.03  | 18.15 | 14.52 |
> | PHE      | 60.33     | 11.80 | 40.26 | 59.85    | 9.04  | 24.61 | 10.15 |
> | Ours     | 64.42     | 14.51 | 46.56 | 63.79    | 11.44 | 27.60 | 8.32  |
> | 800      |           |       |       |          |       |       |       |
> | GMP      | 55.56     | 8.37  | 30.97 | 53.50    | 5.34  | 14.76 | 18.75 |
> | PHE      | 58.35     | 9.18  | 36.20 | 55.03    | 7.31  | 18.12 | 16.00 |
> | Ours     | 62.92     | 12.54 | 43.25 | 60.39    | 10.74 | 25.00 | 10.69 |
>
> Across all scales (200→800 unknown classes), our method outperforms GMP and PHE, indicating that the proposed memory-based discovery mechanism remains stable even when the number of novel categories grows fourfold.
> These results suggest that HM generalizes well to large-scale streams and that its memory-based discovery framework retains robustness even under significantly expanded open-world settings. We will include these results in the revised version. Evaluation on non-vision modalities is promising future work and beyond the scope of this paper.
>
> ### Q1 How to prevent early pseudo-label errors from propagating?
>
> We appreciate the reviewer’s question. This concern precisely aligns with the motivation behind the third component of our framework, i.e., the self-correction mechanism. In HM, prediction confidence is implicitly calibrated through the agreement between the query and the memory structure: confident predictions correspond to consistent nearest-neighbor evidence within the assigned bucket, while uncertain predictions typically exhibit weak or conflicting support across buckets. This consistency-based signal determines whether a new sample should reinforce an existing class or remain tentative.
>
> To prevent early pseudo-label errors from propagating, HM performs periodic self-correction on the memory entries. Samples whose stored labels become inconsistent with the evolving local structure are re-evaluated and reassigned. Because memory retrieval is localized, these corrections remain lightweight while effectively removing mislabeled instances that could otherwise misguide future assignments. As a result, early errors do not accumulate: they are detected as structural inconsistencies and rectified before affecting subsequent discovery. We will clarify this error-prevention mechanism more explicitly in the revision.

---

### Official Review · Reviewer_18vv · 2025-11-01

**Soundness:** 3
**Presentation:** 2
**Contribution:** 3
**Rating:** 4
**Confidence:** 3

**Summary:**

The paper deals with the problem of discovering novel classes in real-time test scenarios. Along with discovering novel samples, the task is also to maintain the performance of known classes. To tackle the challenge of online discovery, the proposed work maintains a hash memory of feature norm and direction. A new class is discovered by querying the buckets. For robustness against noisy data, the top-k neighbors of a bucket are augmented while applying discovery process. For the final decision, global prototypes and hash memory are leveraged to ensure performance in both known and novel classes. Finally, to purify misassigned samples for novel discovered classes, a memory self-correction mechanism is applied. Experiments are conducted in 4 datasets.

**Strengths:**

- Real-time discovery of novel classes is a practical setting of NCD, as introduced by the paper. The problem of TTD is well-motivated by highlighting rediscoveries of novel classes from existing solutions that focus on non-real-time-based postdoc evaluation.
- The paper gives a good overview of the related works regarding NCD and TTT.
- The proposed method is practical for real-time discovery problem.

**Weaknesses:**

- Will the feature norm and directions of the first identified sample of a novel class be representative to identify all the future samples of the same class? As we see more samples, do we need aggregation?
- Some of the details in the paper are missing. For example: How to construct the dynamic graph? What is the frequency of updating the graph?
- The notation k is used for multiple purposes. It is recommended to use distinct notations for specific purposes.
- What is the value of ε, α? How to determine the values?
- If the graph is not used, how does it affect the performance?  Also, how does k in the top-k of the graph neighborhood impact the performance?
- How to determine the optimal size of stored samples in the self-correction module?

**Questions:**

Please refer to the weakness section

---

> ### Author Response · Authors · 2025-11-23
>
> ### W1 Representativeness of the first novel-class sample
>
> We appreciate the reviewer’s insightful question. Indeed, the first discovered sample of a novel class may not always be fully representative. This is precisely why HM does not rely on a single initial sample to define a class anchor. Instead, we incorporate two mechanisms to progressively stabilize the representation as more samples arrive: (1) a self-correction process that revisits early assignments and rectifies possible mistakes, and (2) a controlled EMA-based update of novel-class prototypes to aggregate information gradually without amplifying early noise. These components ensure that class prototypes become more representative over time, and that early misidentifications do not propagate. Our experiments confirm that these refinement steps are essential for stable discovery under TTD (Table 7 (updated 8) and Fig. 5).
>
> ### W2 How to construct the dynamic graph? and the update frequency.
>
> We thank the reviewer for pointing this out. In our method, each hash bucket is treated as a node, and we construct a dynamic graph by linking buckets based on the cosine similarity of their mean feature directions. The graph is dynamic because the content of each bucket evolves as the memory buffer receives new samples, which continuously updates the mean direction of each bucket. Consequently, the bucket-to-bucket similarity and edge structure are refreshed automatically after each memory update. This event-driven update is lightweight and ensures that the graph always reflects the current structure of the streaming data.
>
> ### W3 The notation k is used for multiple purposes
>
> We thank the reviewer for highlighting the ambiguity in our notation. In the current draft, $\kappa$ denotes the norm discretization coefficient, $k$ refers to the number of compared neighbor samples in the bucket graph, and $K$ is the per-class memory capacity. We agree that reusing similar symbols may cause confusion. In the revision, we will assign clearly differentiated notations to these three quantities to ensure unambiguous and consistent usage throughout the paper.
>
> ### W4 What is the value of $\epsilon$ and $\alpha$? How to determine the values?
>
> We thank the reviewer for the question. Both $\epsilon$ and $\alpha$ are analyzed in the appendix.
>
> **$\epsilon$ (Appendix E.2, Fig. 11):**
> $\epsilon$ is the global-to-local confidence boundary deciding whether a sample is classified by the prototype classifier or the LSH-based classifier. As shown in Fig. 12, increasing $\epsilon$ favors the LSH classifier and sharply reduces KA, while decreasing $\epsilon$ overly favors the prototype classifier and destabilizes novel class discovery. A moderate boundary yields the most reliable trade-off between known-class accuracy and new-class discovery.
>
> **$\alpha$ (Appendix D.4, Tables 10 and 11 (updated 11 and 12)):**
> $\alpha$ controls the EMA update of prototypes. Our experiments in the tables show that overly aggressive updates ($\alpha$ close to 1) amplify early discovery errors and reduce TA, while very small $\alpha$ under-adapts. A moderate EMA (around $\alpha \approx 0.9$) provides the most stable performance. For known classes, we keep prototypes fixed to avoid forgetting and reduce confusion.
>
> ### W5 How if without graph? How the k in the top-k of the graph neighborhood impact the performance?
>
> We thank the reviewer for the question. The case of “not using the graph” corresponds to disabling bucket-to-bucket neighbor augmentation, i.e., using only the samples within the same hash bucket. This ablation is already included in **Appendix D.1 (Fig. 9)**, where $k=0$ denotes the no-graph setting. As shown in the figure, removing the graph reduces discovery accuracy because the model receives fewer informative neighbors, confirming that the graph acts as a lightweight stabilizer rather than a critical dependency.
>
> Regarding the choice of $k$, Appendix D.1 further shows a clear trade-off: very small $k$ provides insufficient reference information, while very large $k$ introduces noisy neighbors and increases computational cost. A moderate range achieves the best balance between accuracy and efficiency in the streaming TTD setting.

---

> ### Author Response · Authors · 2025-11-23
>
> ### W6 Optimal size of stored samples?
>
> We thank the reviewer for raising this question. The self-correction (SC) module does not maintain a separate buffer. Instead, it operates directly on the samples stored in the class-wise memory. Therefore, the “size of stored samples in SC” corresponds to the per-class memory capacity $K$. This relationship is analyzed in **Table 6 (updated 7)** of the main paper.
> As shown in the table, increasing $K$ provides more evidence for SC and improves cluster agreement (CA), but too large a buffer accumulates early pseudo-label noise and reduces TA. Very small $K$ makes SC unstable due to insufficient reference samples. A moderate memory size (around 10–20 samples per class) yields the best balance for SC effectiveness and overall discovery stability. This demonstrates that SC does not rely on sensitive tuning, and that the optimal range naturally emerges from the trade-off captured in the memory-size sweep.

---

### Official Review · Reviewer_1TV7 · 2025-11-01

**Soundness:** 3
**Presentation:** 2
**Contribution:** 3
**Rating:** 4
**Confidence:** 4

**Summary:**

The paper introduces Test-Time Discovery (TTD), a real-time evaluation framework for novel class discovery, along with a training-free Hash Memory (HM) method that integrates semantic-aware hashing, a global-to-local prototype-to-LSH classifier, and a lightweight self-correction mechanism. TTD focuses on per-sample decision-making during streaming evaluation rather than relying on post-hoc clustering. Experiments on CIFAR100D, CUB-200D, Tiny-ImageNetD, and AircraftD demonstrate consistent improvements on new real-time metrics (TA and CA) while maintaining stability on known classes.

**Strengths:**

- The TTD protocol effectively reveals practical issues often overlooked in post-hoc NCD and clearly identifies three key challenges.
- The method is appealingly simple and efficient at test time. Hash codes represent both feature norm and direction, LSH buckets are reused to avoid redundant rediscovery, and known-class predictions rely on a fallback prototype classifier.
- Overall, the paper is well written and easy to follow.

**Weaknesses:**

- The semantic-aware hash employs random projections and a norm bin, which is practical but offers limited conceptual novelty compared with existing prototypical hashing or LSH-based retrieval methods. Please provide a clearer positioning against closely related online discovery and hashing approaches to highlight the contribution.
- The approach relies on several design choices, such as κ for norm binning, the number of random directions, the number of bucket-graph neighbors (k), memory size (K), EMA factor (α), and the self-correction cadence. The current sensitivity analysis is limited; robustness claims would be more convincing with systematic parameter sweeps and a clearer examination of latency–memory trade-offs under streaming conditions.

**Questions:**

- How were κ (norm discretization), number of random directions n, bucket neighbor count k, and memory size K chosen? Please provide ranges, validation protocols, and wall-clock latency/throughput under streaming loads.
- Have you tried larger-scale open-world streams (e.g., ImageNet-derived streams with >1k categories) or non-vision modalities?
- Dataset split descriptions are useful; a quick table in the main text (with known/unknown counts and stream length) would aid readability.

---

> ### Author Response · Authors · 2025-11-23
>
> ### W1 Conceptual positioning of the hashing module
>
> TTD requires **discovery and testing to occur simultaneously**, where each prediction must be made *immediately* and cannot be revised later. This online, non-revisitable setting introduces stability and identity-preservation challenges that do not appear in post-hoc NCD/GCD/OCD evaluations.
>
> Existing online or on-the-fly discovery methods (e.g., PHE) operate online but are **tested and evaluated post-hoc**. They do not require a mechanism to maintain persistent class identity through test time. Under TTD, the absence of such a mechanism directly leads to redundant rediscovery and error accumulation.
>
> In this context, the hashing used in HM is **not intended as a new hashing technique**. It is a lightweight way to partition the memory for efficient lookup in streaming conditions. Our goal is not to improve hashing itself, but to enable a **memory-based discovery and correction mechanism** that remains stable throughout TTD.
>
> Thus, hashing serves as an efficient organizational tool within the memory framework, while the contribution lies in the design of this TTD-oriented memory mechanism, instead of the hashing operation itself.
>
> ### W2&Q1 Hyperparameter selection and their streaming-time efficiency impacts
>
> We thank the reviewer for highlighting the concern regarding design choices and sensitivity analysis.
> We would like to clarify that the selections of these hyperparameters have already been analyzed in the appendix or main paper:
>
> - **Norm discretization $\kappa$ (Appendix Table 12 (updated 13)):** $\kappa$ shows a predictable trade-off: large values over-amplify norm differences and hurt KA, whereas small values weaken cluster alignment. A moderate $\kappa$ achieves the most stable behavior.
> - **Random direction count (Appendix D.3, Fig. 10):** Too few directions fail to sufficiently partition the angular space, and too many introduce unnecessary bucket fragmentation and search cost. A mid-range choice offers the best balance between effectiveness and efficiency.
> - **Number of compared samples $k$ (Appendix D.2, Table 9 (updated 10)):** Very small $k$ limits useful reference information, while very large $k$ brings in noisy neighbors. Intermediate values yield the most reliable discovery performance.
> - **Memory size $K$ (Main Table 6 (updated 7)):** Increasing $K$ strengthens cluster agreement but also accumulates early pseudo-label noise, lowering TA. Moderate memory sizes strike the best balance for stable online discovery.
> - **EMA factor $\alpha$ (Appendix D.4):** Updating prototypes too aggressively propagates early errors, whereas overly slow updates fail to track new classes. A moderate EMA provides the most consistent results.
> - **Self-correction cadence (Main Fig. 6):** Very frequent corrections raise computation without clear benefit, while very sparse corrections let errors accumulate. A balanced cadence yields the most robust behavior under TTD.
>
> For latency–memory trade-offs under streaming conditions, our additional measurements show that HM’s runtime and memory usage scale smoothly with all hyperparameters, including memory size $K$, norm discretization $\kappa$, bucket neighbors, and top-$k$ comparisons. Across all settings, the changes in latency follow predictable linear trends and memory consumption remains effectively constant. No configuration introduces abrupt slowdowns or nonlinear growth. These results confirm that the proposed components impose only mild, controllable overhead, and that HM maintains efficient real-time operation throughout the TTD stream.
>
>
> | stored samples/class | Time(s) | MB   |
> | -------------------- | ------- | ---- |
> | 0                    | 484.24  | 8799 |
> | 10                   | 495.51  | 8831 |
> | 20                   | 505.55  | 8863 |
> | 30                   | 521.59  | 8895 |
>
> | $\kappa$ | Time(s) | MB   |
> | -------- | ------- | ---- |
> | 1        | 552.14  | 8865 |
> | 2        | 505.55  | 8863 |
> | 4        | 481.17  | 8863 |
> | 10       | 486.84  | 8862 |
>
> | #bucket | Time(s) | MB   |
> | ------- | ------- | ---- |
> | 0       | 362.38  | 8863 |
> | 1       | 424.53  | 8863 |
> | 2       | 489.73  | 8863 |
> | 3       | 505.55  | 8863 |
> | 4       | 536.21  | 8863 |
> | 5       | 561.46  | 8863 |
> | 6       | 566.45  | 8863 |
>
> | Top-$k$ compared samples | Time(s) | MB   |
> | ------------------------ | ------- | ---- |
> | 0                        | 452.60  | 8863 |
> | 5                        | 483.45  | 8863 |
> | 10                       | 505.55  | 8863 |
> | 15                       | 520.11  | 8863 |
> | 20                       | 532.74  | 8863 |

---

> ### Author Response · Authors · 2025-11-23
>
> ### Q2 Larger-scale open-world streams
>
>
> We thank the reviewer for raising this important question about scalability.
> To evaluate HM under larger-scale open-world conditions, we conducted additional experiments using an ImageNet-derived stream with more than 1,000 categories. To avoid retraining large models and focus specifically on test-time scalability, we used a model pretrained on Tiny-ImageNet (200 classes) and evaluated it on the full ImageNet-resized test set (200 + 800 unknown), varying the number of unknown categories from 200 to 800.
> The results are shown below:
>
> | #Unknown | REAL-TIME |       |       | POST-HOC |       |       |       |
> | -------- | --------- | ----- | ----- | -------- | ----- | ----- | ----- |
> | 200      | KA        | TA    | CA    | KA       | TA    | CA    | KF    |
> | GMP      | 65.67     | 16.14 | 46.57 | 64.80    | 10.88 | 27.10 | 10.50 |
> | PHE      | 67.75     | 16.29 | 50.94 | 66.27    | 13.24 | 28.61 | 7.59  |
> | Ours     | 70.93     | 18.47 | 55.78 | 71.28    | 16.07 | 33.74 | 3.62  |
> | 400      |           |       |       |          |       |       |       |
> | GMP      | 63.58     | 13.41 | 45.76 | 61.15    | 7.81  | 20.54 | 11.36 |
> | PHE      | 65.11     | 14.19 | 47.32 | 64.53    | 9.63  | 24.23 | 8.26  |
> | Ours     | 68.09     | 16.77 | 51.85 | 67.50    | 14.31 | 30.16 | 5.86  |
> | 600      |           |       |       |          |       |       |       |
> | GMP      | 58.49     | 10.29 | 37.69 | 57.98    | 6.03  | 18.15 | 14.52 |
> | PHE      | 60.33     | 11.80 | 40.26 | 59.85    | 9.04  | 24.61 | 10.15 |
> | Ours     | 64.42     | 14.51 | 46.56 | 63.79    | 11.44 | 27.60 | 8.32  |
> | 800      |           |       |       |          |       |       |       |
> | GMP      | 55.56     | 8.37  | 30.97 | 53.50    | 5.34  | 14.76 | 18.75 |
> | PHE      | 58.35     | 9.18  | 36.20 | 55.03    | 7.31  | 18.12 | 16.00 |
> | Ours     | 62.92     | 12.54 | 43.25 | 60.39    | 10.74 | 25.00 | 10.69 |
>
> Across all scales (200 to 800 unknown classes), our method outperforms GMP and PHE, indicating that the proposed memory-based discovery mechanism remains stable even when the number of novel categories grows fourfold.
> These results suggest that HM generalizes well to large-scale streams and that its memory-based discovery framework retains robustness even under significantly expanded open-world settings. We will include these results in our paper.
>
>
>
> ### Q3 A quick table for dataset split descriptions
>
> We thank the reviewer for the suggestion. The full dataset construction and class-split statistics (known/unknown counts and total stream size) are already provided in **Appendix Section B and Table 8  (updated 9)**. To improve clarity, we have included a compact summary table (updated Table 2) in the main text that reports the number of known/unknown classes for each dataset, so readers can quickly access the key split information without consulting the appendix.

---

### Author Response · Authors · 2025-11-28
**Revision Summary**

Dear Reviewers:

Thank you for the valuable comments. In the revised manuscript, we have incorporated the requested clarifications and resolved the remaining ambiguities. Revisions marked in red highlight the major changes. Below, we summarize the modifications made in response to the reviewer’s comments:

1.	Added a clarification at the end of Sec. 4.2 explaining that hashing is introduced to efficiently organize the memory structure.
2.	Added a unified runtime and memory table (varying key hyperparameters) in Appendix E.6.
3.	Added a quick table in Sec. 5.1 summarizing known and unknown class counts, and clarified stream length (batch size 50) in Appendix B.
4.	Clarified in Sec. 4.3 why the graph structure is dynamic in the TTD setting.
5.	Revised the beginning of Introduction (second paragraph) to clearly state that TTD performs test-time prediction and class discovery simultaneously with real-time evaluation.
6.	Strengthened Sec. 3.1 to highlight the distinction between post-hoc evaluation and real-time test-time evaluation.
7.	Clarified in Introduction (second paragraph) how TTD differs from traditional NCD post-hoc evaluation.
8.	Added the missing PHE citation and description in the OCD part of Related Work.
9.	Revised the GCD discussion in Related Work to emphasize that GCD requires full-access global clustering over the entire test set.
10.	Added an explanation in Appendix B for choosing the initial 7:3 known–unknown ratio.
11.	Added Stanford Cars and ImageNet-resized (1000-class) experimental results in Appendix F.
12.	Added a justification in Appendix C explaining why classical post-hoc metrics (e.g., GCD metrics) are not applicable to TTD.
13.	Clarified in Sec. 5.1 that strong performance requires jointly achieving good TA and CA.
14.	Clarified in Sec. 5.4 that the first row of Table 7 corresponds to a thresholding baseline with EMA-updated prototypes.

We hope these revisions fully resolve the reviewer’s questions and improve the clarity and rigor of the paper.

Best regards,

The authors

---

### Author Response · Authors · 2025-12-02
**To the new AC**

Dear AC,

Thank you for your attention to our submission. Given the transition to a new AC, we provide a concise summary of the paper’s contributions, the reviewers’ main concerns, and how our rebuttal and revised manuscript address them.

## Core Contributions and Novelty

Our work introduces the Test Time Discovery setting, where a model must classify known categories and discover new ones within the same streaming test sequence under real-time constraints. This setting differs from existing NCD or GCD that separate discovery and testing and rely on full-set test data access and post-hoc clustering. We propose a Hash Memory framework that performs discovery and prediction simultaneously through a memory mechanism that preserves class identity online. The contribution lies in the memory organization and the online prototype refinement designed specifically for the TTD constraints.

## Key Reviewer Concerns and Our Responses

1. **Distinction between TTD and GCD/OCD/post hoc NCD**: Reviewers (1TV7 and ucGd) asked whether TTD is meaningfully different from existing settings.

**Response**: **All existing discovery settings separate discovery and testing and rely on full-set access or post-hoc clustering**. They do not operate in a streaming test regime. To our knowledge, **TTD is the first work where discovery and prediction occur simultaneously at test time**, with per-sample irreversible decisions and no visibility of future data, which is essential in *real-world scenarios such as autonomous driving which must make immediate per-sample decisions for both prediction and discovery*. We also showed that GCD collapses when forced into this regime (requested by ucGd), confirming the necessity and uniqueness of TTD.

2. **Methodological clarity and the role of hashing**: Reviewers (1TV7, 18vv and vbHh) asked whether hashing is itself a contribution and whether the method is conceptually well defined.

**Response**: We clarified that the method has three tightly coupled components specifically required by TTD: (1) a memory that organizes features into stable buckets, (2) prediction and discovery via efficient local retrieval within memory buckets, and (3) lightweight self correction to avoid early error propagation. Hashing is not claimed as a novelty but is simply an efficient mechanism to organize the memory. **The contribution is the memory based TTD mechanism**, not the hashing operation, and this is now made explicit.

3. **Sensitivity of key hyperparameters**: Reviewers (1TV7, 18vv and ucGd) comments that the manuscript should include hyperparameter experiments.

**Response**: We showed that the key hyperparameters raised by the reviewers **were already analyzed in the main paper or appendix**. In the rebuttal we *redirected* reviewers to the exact tables and figures and supplemented them with additional runtime and memory measurements. These collectively confirm that the method behaves predictably and is robust across parameter ranges.

4.	**Necessity and definition of real time evaluation**: Reviewers (ucGd) questioned why new evaluation metrics are needed and whether real-time evaluation is justified.

**Response**: We clarified that TTD requires irreversible, per sample decisions on a streaming test set. Classical NCD/GCD/OCD metrics rely on post hoc clustering of the full dataset, which cannot be computed in TTD without violating the setting itself. Our agreement metrics are the natural real time equivalents of those measures and evaluate discovery and classification as they happen. This resolves the conceptual ambiguity and explains why TTD necessarily departs from post hoc evaluation.

5.	**Scalability to large-class streams**: Reviewers (1TV7, vbHh and ucGd) asked whether the method would scale on larger-class benchmarks.

**Response**: We added extensive large-scale experiments on Tiny-ImageNet + 200 to 800 ImageNet-resized unknown classes (1,000 classes in total). Our method remains stable and consistently outperforms GMP and PHE, demonstrating strong scalability and confirming that the memory-based design is effective even under large class spaces.

## Summary

The only reviewer who followed up (ucGd) explicitly stated that our additional experiments and clarifications fully addressed his/her concerns regarding prior work, GCD comparison, known–unknown ratios, dataset choices, and missing ablations. The only remaining question concerned an inconsistency in Table 7, which we resolved by unifying the KF definition and clarifying the EMA baseline. Therefore, we believe, with these corrections, all issues raised across reviews have been settled.

**All updates are highlighted in red in the revised manuscript, as we summarize in another comment**. The contribution is now clearly presented, rigorously supported, and distinct from existing discovery settings. We respectfully ask the Area Chair to evaluate the manuscript and rebuttal holistically.

Sincerely,
The authors

---

### Meta-Review · Area_Chair_397p · 2025-12-31

**Summary:**

This paper received mixed initial ratings (2, 4, 4, 6). The main concerns can be summarized as follows: (i) contribution positioning and sensitivity of the semantic-aware hash [1TV7, 18vv]; (ii) sensitivity analysis and the individual contributions of various design choices [1TV7, 18vv, vbHh]; (iii) details of and the usage of the dynamic graph [18vv]; (iv) ambiguity in the real-time evaluation [vbHh]; (v) unclear relationships to, and distinguishing contributions from, prior work (e.g., OCD, GCD) [ucGd]; (vi) experimental design and fairness (dataset splits, evaluation protocols, etc.) [ucGd]; and (vii) limited ablation studies [ucGd].

The AC has carefully read the revised paper, the reviews, and the author responses. The authors criticize prior settings in GCD, NCD, and OCD, arguing that discovery and testing are processed separately. They place strong emphasis on “real-time evaluation under the test-time discovery” setting, both in the revised manuscript and in the rebuttal, claiming that this is the first work to simultaneously discover and classify data in a streaming scenario. However, this claim is overstated, and the criticism of OCD is inaccurate. OCD already operates in a streaming testing scenario, where discovery and classification occur simultaneously, and inference on current samples is not affected by future samples. Moreover, OCD is instance-based, which introduces additional challenges.

The paper lacks a thorough discussion of existing OCD settings and approaches. Without such clarification, the proposed new dataset splits and evaluation metrics appear largely redundant compared to OCD [ucGd]. In addition, as shown in the ablation tables, the inconsistent performance across different evaluation metrics further raises concerns about the effectiveness of the proposed metrics. Finally, it remains unclear how the proposed method specifically contributes to or benefits this online setting.

The author response regarding the positioning of the hashing technique [1TV7, 18vv] is weak, particularly given the statement that it is “not intended as a new hashing technique.” The same challenges of the online setting (e.g., stability) also arise in OCD, and hashing techniques have already been directly employed in OCD and PHE, which further weakens the claimed novelty of the paper. In addition, memory-based techniques have been widely adopted in test-time adaptation settings, yet related discussions are missing. Overall, the positioning of the paper remains unclear, as its primary claim regarding the setting is weakened, and the proposed method does not convincingly demonstrate design choices that are specifically tailored to that setting.

**Reviewer Concerns:**

Some of the limitations listed in the Summary section were adequately addressed in the author rebuttal. However, the major concern regarding the positioning of the methodology and the clarity of its distinguishing contributions remains unresolved.

**Reviewer Scores:**

The concerns raised by [1TV7, 18vv, vbHh] may be partially addressed; however, additional concerns emerge, as the provided component-wise analyses and sensitivity studies show large fluctuations. As a result, the stand-alone contributions of individual design choices are difficult to determine. Reviewer [ucGd] noted that most concerns, including comparisons with other research lines (e.g., GCD), have been addressed and therefore indicated a possible rating increase to 4. However, the AC believes that a major related concern has been overlooked, which makes some of the author responses less convincing.

---

### Decision · Program_Chairs · 2026-01-26

Reject